# Antagonistic roles for Ataxin-2 structured and disordered domains in RNP condensation

Amanjot Singh[1†], Joern Hulsmeier[2†], Arvind Reddy Kandi[1,3], Sai Shruti Pothapragada[1], Jens Hillebrand[2], Arnas Petrauskas[2], Khushboo Agrawal[3,4], Krishnan RT[1], Devasena Thiagarajan[1], Deepa Jayaprakashappa[1], K VijayRaghavan[1], Mani Ramaswami[1,2]*, Baskar Bakthavachalu[1,3,5]*

[1]National Centre for Biological Sciences, Bangalore, India; [2]Trinity College Institute of Neuroscience, School of Genetics and Microbiology, Smurfit Institute of Genetics and School of Natural Sciences, Trinity College Dublin, Dublin, Ireland; [3]Tata Institute for Genetics and Society Centre at inStem, Bellary Road, Bangalore, India; [4]School of Biotechnology, Amrita Vishwa Vidyapeetham University, Kollam, India; [5]School of Basic Sciences, Indian Institute of Technology, Mandi, India

*For correspondence:
ramaswam@tcd.ie (MR);
baskarb1@gmail.com (BB)

†These authors contributed equally to this work

**Abstract** Ataxin-2 (Atx2) is a translational control molecule mutated in spinocerebellar ataxia type II and amyotrophic lateral sclerosis. While intrinsically disordered domains (IDRs) of Atx2 facilitate mRNP condensation into granules, how IDRs work with structured domains to enable positive and negative regulation of target mRNAs remains unclear. Using the Targets of RNA-Binding Proteins Identified by Editing technology, we identified an extensive data set of Atx2-target mRNAs in the *Drosophila* brain and S2 cells. Atx2 interactions with AU-rich elements in 3′UTRs appear to modulate stability/turnover of a large fraction of these target mRNAs. Further genomic and cell biological analyses of Atx2 domain deletions demonstrate that Atx2 (1) interacts closely with target mRNAs within mRNP granules, (2) contains distinct protein domains that drive or oppose RNP-granule assembly, and (3) has additional essential roles outside of mRNP granules. These findings increase the understanding of neuronal translational control mechanisms and inform strategies for Atx2-based interventions under development for neurodegenerative disease.

## Introduction

Ataxin-2's involvement in human disease, its relevance for therapeutics development, and its established roles in ribonucleoprotein (RNP)-phase transitions, cell physiology, metabolic control, and animal behavior have led to considerable interest in understanding molecular mechanisms by which the protein functions. At a molecular level, Ataxin-2 positively or negatively regulates the translation of specific mRNAs (*Lee et al., 2017*; *Lim and Allada, 2013*; *McCann et al., 2011*; *Zhang et al., 2013*). At the same time, the protein mediates the assembly of mRNPs into cytoplasmic mRNP granules visible in resting neurons or in RNA stress granules (SGs) that occur in most cells in response to stress (*Bakthavachalu et al., 2018*; *Bentmann et al., 2013*). At a cellular level, Ataxin-2 contributes to cell viability and differentiation as well as cellular responses to viral, ER, heat-, and oxidative stress (*Bonenfant et al., 2019*; *Del Castillo et al., 2019*; *Ralser et al., 2005*; *van de Loo et al., 2009*). Finally, at an organismal level, the protein regulates metabolism, circadian rhythm, and the consolidation of long-term memory (*Bakthavachalu et al., 2018*; *Lim and Allada, 2013*; *Meierhofer et al., 2016*; *Pfeffer et al., 2017*; *Zhang et al., 2013*). Parallel clinical genetic studies have shown that genetic mutations in human Ataxin-2 (Atxn2) can cause the hereditary neurodegenerative diseases

spinocerebellar ataxia type II or amyotrophic lateral sclerosis (ALS) (*Daoud et al., 2011*; *Elden et al., 2010*; *Lastres-Becker et al., 2008*; *Lee et al., 2011*; *Scoles and Pulst, 2018*; *Wadia, 1977*; *Wadia and Swami, 1971*), and subsequent work showing that genetic reduction of Ataxin-2 activity slows neurodegeneration in animal models of ALS has inspired the design and development of therapeutics targeting human Ataxin-2 (*Becker et al., 2017*; *Becker and Gitler, 2018*; *Elden et al., 2010*; *Scoles et al., 2017*).

The above biological and clinical studies of Ataxin-2 are connected by the insight that intrinsically disordered domains (IDRs) present on RNA-binding proteins contribute to macromolecular condensation or liquid–liquid phase separation reactions, wherein monomeric units form dynamic assemblies held together by weak multivalent interactions (*Jain and Vale, 2017*; *Kato and McKnight, 2018*; *Murray et al., 2017*; *Saha and Hyman, 2017*; *Van Treeck et al., 2018*; *Van Treeck and Parker, 2018*). Significantly, IDRs not only support assembly of mRNP granules but also are prone to assemble into amyloid-like fibers. Disease-causing mutations often increase the efficiency of amyloid formation, particularly within mRNP granules where the RNA-binding proteins are concentrated (*Courchaine et al., 2016*; *Kato et al., 2012*; *Lim and Allada, 2013*; *Nedelsky and Taylor, 2019*; *Patel et al., 2015*; *Ramaswami et al., 2013*; *Xiang et al., 2015*; *Yang et al., 2019*). The broad proposal that SGs serve as 'crucibles' for the initiation of neurodegenerative disease (*Li et al., 2013*; *Ramaswami et al., 2013*; *Wolozin and Ivanov, 2019*) explains why SG proteins are often mutated in familial ALS or frontotemporal dementia (FTD) and why these proteins are observed in intracellular inclusions typical of ALS/FTD (*Arai et al., 1992*; *Brettschneider et al., 2015*; *Lee et al., 1991*).

The domain structure of Ataxin-2 is highly conserved across species, with N-terminal Like-Sm (Lsm) and Lsm-associated (LsmAD) domains, a more carboxy-terminal polyA binding protein interaction motif 2 (PAM2) domain, as well as strongly disordered regions (respectively mIDR and cIDR) in the middle and C-terminal regions of the protein (*Albrecht et al., 2004*, *Satterfield and Pallanck, 2006*; *Nonhoff et al., 2007*; *Bakthavachalu et al., 2018*). Genetic studies in *Drosophila*, which has a single gene for Ataxin-2 as against two *atxn2* and *atxn2-like* in mammals, indicate that different Atx2 domains encode distinct, biological functions. Specifically, while each structured domain is essential for normal viability, the IDR domains are not. However, the cIDR is required for normal mRNP assembly and long-term memory as well as for facilitating cytotoxicity in *Drosophila* Fus and C9orf72 models for ALS/FTD (*Bakthavachalu et al., 2018*). These observations, while instructive in terms of functions of the Atx2-IDR and mRNP granules, provide no direct insight into other functions and mechanisms mediated by structured domains of Atx2 or their roles in biology.

To better understand the broad roles and mechanisms of Atx2, we used the Targets of RNA-Binding Proteins Identified by Editing (TRIBE) technology (*Biswas et al., 2020*; *McMahon et al., 2016*) to globally identify Atx2-interacting mRNAs from *Drosophila* adult brain and study how these in vivo interactions were influenced by different domains of the protein. In addition to identifying biologically important targets of Atx2, the results described here offer unexpected information into the mechanisms of Atx2 protein function. Atx2 associates with mRNAs predominantly within mRNP granules, where it binds preferentially near AU-rich elements (AREs) in the 3'UTRs to stabilize the majority of the targets. While the cIDR enables mRNA interactions and granule assembly, the Lsm domain reduces both mRNP assembly and Atx2-target interactions. Taken together, our data (1) provide a rich data set of Atx2-target mRNAs, (2) point to a novel essential function of Atx2 outside of mRNP granules, and (3) indicate competing disassembly and pro-assembly activities within Atx2 encoded by the Lsm and IDR domains, respectively. In addition to being of specific biological interest, these conclusions are relevant to current therapeutic strategies based on targeting human Atxn2.

## Results

### Using TRIBE to identify Atx2-target mRNAs in *Drosophila* brain

Atx2 is abundantly expressed in brain tissue. To identify in vivo targets of Atx2 in *Drosophila melanogaster* brain, we used TRIBE, a technology previously shown to reproducibly identify RNA binding protein (RBP) target mRNAs in vivo (*McMahon et al., 2016*). We generated transgenic flies that express Atx2 linked to the catalytic domain of *Drosophila* RNA-modifying enzyme adenosine deaminase (ADARcd) at the carboxy terminal along with a V5 epitope tag under the control of the Gal4-

responsive UAS promoter (*Figure 1—figure supplement 1*). In tissue expressing the Atx2-ADARcd transgene, mRNAs should undergo adenosine-to-inosine editing specifically at positions proximal to Atx2 binding sites (*Figure 1A*). In an *elav-Gal4* background, where the Gal4 transcription factor is expressed in postmitotic neurons, Atx2-ADARcd is expressed specifically in the nervous system. We further temporally restricted neural expression to adult flies with the use of the temperature-sensitive Gal4 inhibitor, GAL80$^{ts}$, that is active at temperatures below 18°C. Thus, in *elav-Gal4; Tub-Gal80$^{ts}$, UAS-Atx2-ADARcd* adult flies shifted from 18°C to 29°C for 5 days shortly after eclosion, and neural mRNAs expressed in adult flies would be susceptible to editing at adenosine residues proximal to Atx2 binding sites.

To identify neural mRNA targets of Atx2, we isolated polyA-selected RNAs from Atx2-ADARcd expressing *Drosophila* brain and sequenced these using Illumina Hiseq 2500 and reads were subsequently analyzed according to *McMahon et al., 2016* (with slight modifications, see Materials and methods) to identify sites and efficiency of Atx2-ADARcd-mediated mRNA editing (*Figure 1A*). Adult brain expression for Atx2-ADARcd was verified using antibody staining for V5 epitope (*Figure 1—figure supplement 2*). Experiments were carried out in duplicates. The reads obtained were between 20 and 25 million per sample, were of high quality, and more than 80% of these mapped to specific *Drosophila* mRNAs (*Figure 1—figure supplement 3*).

Transcripts corresponding to 8% of expressed genes showed adenosine edits. Of the 528 and 481 edit sites represented in independent replicates, at least 317 were common to both samples, demonstrating the reproducibility of the experiments (*Figure 1B*). These 317 common edits could be assigned to 256 unique genes, with the majority of genes only being edited at a single site (215 genes, 67.8% of edits) (*Figure 1C*). The Atx2-target mRNAs were distributed across the brain transcriptome irrespective of the abundance of an mRNA indicating that edits were not random events, but rather reflect sequence- and structure-specific association of these mRNAs with Atx2-ADARcd (*Figure 1D*). We further validated the TRIBE analysis/Illumina Hiseq pipeline using Sanger sequencing for a few identified target mRNAs. For instance, Sanger sequencing confirmed that edits in *Adipokinetic hormone* (*Akh*) and *14-3-3 epsilon* mRNAs occur at identified sites, with efficiencies indicated by TRIBE analysis (*Figure 1E*, *Figure 1—figure supplement 4*). Taken together, the data indicates that Atx2 associates with specific RNA motifs present on targets identified by TRIBE analyses from fly brain.

A Gene Ontology analysis of Atx2 targets (*Table 1*) indicated particular enrichment of mRNAs encoding neuropeptides and hormones, as well as mRNAs encoding monoamine transporters, ion channels, and vesicle transport proteins. This is broadly consistent with and suggests the mechanisms by which Atx2 functions in translational control of physiological and neural circuit plasticity.

## Atx2 associates preferentially with 3′UTRs of target mRNAs

The RNA edit sites identified by TRIBE reflect the positions to which ADARcd is targeted via direct or indirect Atx2–mRNA interactions. It is therefore possible to use this information to determine the relative positions and preferred sequences for Atx2 binding on respective mRNAs. By converting the TRIBE edits into metagene coordinates, we found that Atx2 interactions occurred predominantly within 3′UTRs of target mRNAs (69.5%), while the coding region (CDS) and 5′UTR accounted for 26.7% and 4% of edits, respectively (*Figure 2A*). All the identified edits occurred almost exclusively in exons (*Figure 2—figure supplement 1*), which might be due to polyA selection causing experimental bias or that Atx2 binds to only mature mRNA in the cytoplasm. Edits within CDS were often accompanied by edits in the 3′UTR of the same mRNA, further indicating a key role for Atx2–3′UTR interactions (*Figure 2—figure supplement 2*). Also pointing to a role in 3′UTR regulation, Atx2 edit sites were particularly prevalent in brain transcripts with longer 3′UTRs that are more often subject to translational control (*Inagaki et al., 2020*; *Miura et al., 2013*; *Wang and Yi, 2014*; *Figure 1—figure supplement 3*). These observations are consistent with prior work showing that Atx2 can mediate activation or repression of specific mRNA translation via elements in their 3′UTRs (*Lee et al., 2017*; *Lim and Allada, 2013*; *McCann et al., 2011*; *Sudhakaran et al., 2014*; *Zhang et al., 2013*).

Further analyses using the MEME suite of tools for motif-based sequence analysis identified an AU-rich element (ARE) 'UAUAUAUA' as highly enriched in mRNA target sequences within 100 bases of identified edit sites (*Figure 2B*). These AREs, previously implicated in the regulation of mRNA stability, are most abundant near 3′UTR edits sites of the target mRNAs, while a secondary motif with

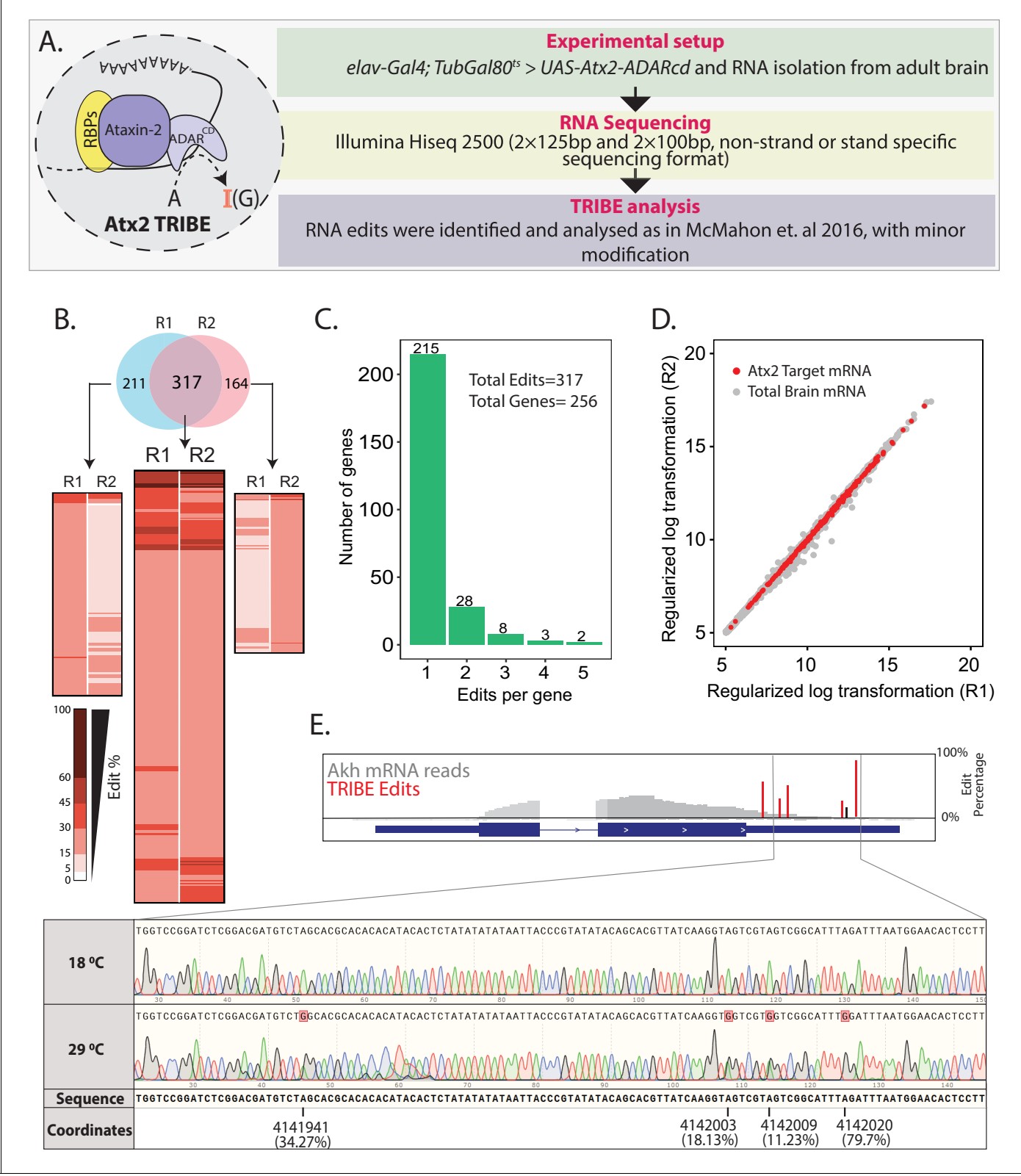

**Figure 1.** Using Targets of RNA-Binding Proteins Identified by Editing (TRIBE) to identify Atx2-interacting mRNAs in the adult fly brain. (**A**) Schematic and flow chart for TRIBE analysis: the Atx2-ADARcd fusion protein is expressed in the adult fly brain, total brain mRNA isolated, sequenced, and analyzed using a published TRIBE pipeline. (**B**) Heatmaps show edit percentages of individual transcript coordinates. Replicate experiments R1 and R2 identify largely overlapping edit sites and edited mRNAs. The common targets (intersect between R1 and R2) show almost reproducible edit levels. The

*Figure 1 continued on next page*

*Figure 1 continued*

inset (heatmaps on either sides of the common intersected list) indicates that several mRNAs are identified as 'non-replicates' between R1 or R2 because they do not cross quality control thresholds (edit percentages or read counts) and not because of robust differences between replicates. (C) Bar plot showing the number of edits with a significant number of genes edited at a single site. (D) The mRNA edits are independent of expression levels. A scatter plot shows expression differences of all the mRNAs expressed in fly brain. Red dots represent the edited mRNAs while gray dots represent the mRNAs of the brain transcriptome that are not edited. (E) Sanger sequencing confirms editing site identified by TRIBE analysis. Data shown for one target mRNA (Akh). Red bars show the edit percentages at the different modified nucleotides with respect to the total Akh mRNA shown in gray. The black bar indicates identified edits that are below 15% threshold (see also *Figure 1—figure supplement 4*).

The online version of this article includes the following source data and figure supplement(s) for figure 1:

**Source data 1.** Data related to *Figure 1B*.
**Source data 2.** Data related to *Figure 1C*.
**Figure supplement 1.** *Atx2-ADARcd* fusion protein constructs.
**Figure supplement 2.** UAS-Atx4-ADARcd expression was driven using *elav-Gal4;tub-Gal80*[ts] in adult fly brain for 5 days.
**Figure supplement 3.** Unique mapping percentage for various RNAseq replicates.
**Figure supplement 4.** An example of a gene, *14-3-3 epsilon*, showing multiple edits specifically in the exons.

**Table 1.** Gene Ontology (GO) analysis of Atx2 brain targets.
GO analysis using PANTHER shows a large enrichment of neuronal mRNAs coding for neuropeptides and proteins involved in neuronal signaling pathways. FDR: false discovery rate.

| Molecular function | Fold enrichment | Probability (FDR) | Targets (%) | Genome (%) | GO term |
|---|---|---|---|---|---|
| Signaling receptor binding | 7.75 | 4.47E-11 | 14.46 | 1.87 | GO:0005102 |
| Neuropeptide hormone activity | 27.67 | 2.52E-09 | 6.63 | 0.24 | GO:0005184 |
| Hormone activity | 18.64 | 5.74E-08 | 6.63 | 0.36 | GO:0005179 |
| Neuropeptide receptor binding | 38.74 | 2.08E-06 | 4.22 | 0.11 | GO:0071855 |
| G protein-coupled receptor binding | 13.84 | 3.79E-06 | 6.03 | 0.44 | GO:0001664 |
| Receptor regulator activity | 9.58 | 4.55E-06 | 7.23 | 0.76 | GO:0030545 |
| Receptor ligand activity | 9.41 | 1.96E-05 | 6.63 | 0.71 | GO:0048018 |
| Signaling receptor activator activity | 9.04 | 2.58E-05 | 6.63 | 0.74 | GO:0030546 |
| Organic hydroxy compound transmembrane transporter activity | 83.02 | 3.38E-04 | 2.41 | 0.03 | GO:1901618 |
| Monoamine transmembrane transporter activity | 66.41 | 5.52E-04 | 2.41 | 0.04 | GO:0008504 |
| Phosphoric ester hydrolase activity | 4.93 | 6.43E-04 | 7.84 | 1.59 | GO:0042578 |
| Anion:sodium symporter activity | 55.35 | 6.44E-04 | 2.41 | 0.05 | GO:0015373 |
| Sodium:chloride symporter activity | 55.35 | 6.84E-04 | 2.41 | 0.05 | GO:0015378 |
| Phosphoprotein phosphatase activity | 7.95 | 7.06E-04 | 5.43 | 0.69 | GO:0004721 |
| Anion:cation symporter activity | 36.9 | 1.81E-03 | 2.41 | 0.07 | GO:0015296 |
| Cation:chloride symporter activity | 36.9 | 1.91E-03 | 2.41 | 0.07 | GO:0015377 |
| Phosphatase activity | 5.05 | 2.53E-03 | 6.63 | 1.32 | GO:0016791 |
| Serotonin:sodium symporter activity | 83.02 | 4.17E-03 | 1.81 | 0.03 | GO:0005335 |
| Neurotransmitter:sodium symporter activity | 14.82 | 5.04E-03 | 3.02 | 0.21 | GO:0005328 |
| Protein tyrosine/serine/threonine phosphatase activity | 23.72 | 6.17E-03 | 2.41 | 0.11 | GO:0008138 |
| Solute:sodium symporter activity | 11.53 | 1.26E-02 | 3.02 | 0.27 | GO:0015370 |
| mRNA 3'UTR binding | 11.22 | 1.36E-02 | 3.02 | 0.27 | GO:0003730 |
| Protein tyrosine phosphatase activity | 10.64 | 1.59E-02 | 3.02 | 0.29 | GO:0004725 |
| Neurotransmitter transmembrane transporter activity | 10.64 | 1.65E-02 | 3.02 | 0.29 | GO:0005326 |
| Adrenergic receptor activity | 24.91 | 3.98E-02 | 1.81 | 0.08 | GO:0004935 |

The online version of this article includes the following source data for Table 1:
**Source data 1.** Gene Ontology (GO) terms for Atx2 brain targets.

GC-rich sequence was found predominantly for the CDS edits (*Figure 2—figure supplement 4*). A multiple-sequence alignment for two of the genes (*Vmat* and *Akh*) shows the motifs in the 3′UTR are conserved between closely related *Drosophila* species (*Figure 3—figure supplement 5*). This suggests a model in which Atx2 preferentially associates with ARE-containing 3′UTRs of target mRNAs to regulate their stability in vivo either by directly binding the target mRNA or indirectly via another ARE-binding protein.

## Atx2 stabilizes the majority of its mRNA targets

AREs are major *cis*-regulatory motifs in the 3′UTR of mRNAs that regulate their stability (*Otsuka et al., 2019*; *Vasudevan and Peltz, 2001*). For this reason, several RBPs modulate mRNA stability by binding to AREs and regulating their accessibility to RNA degradative machinery (*Mayya and Duchaine, 2019*). For example, Pumilio binding to ARE increases degradation, while HuR binding stabilizes the mRNAs (*López de Silanes et al., 2004*; *Weidmann et al., 2014*). To ask how Atx2 binding alters target mRNA stability, we expressed a previously validated Atx2-targeting RNAi construct in fly brains to reduce endogenous Atx2 expression and used RNAseq to determine how this affected the steady-state levels of Atx2-target and non-target mRNAs (*McCann et al., 2011*; *Sudhakaran et al., 2014*).

Experimental *elav-Gal4, UAS-Atx2-RNAi; Tub-Gal80$^{ts}$* flies were reared to adulthood at 18°C. They were then transferred to 29°C to inactivate Gal80$^{ts}$ and enable neural Atx2-RNAi expression for 5 days before isolating total RNA from brain (*Figure 3A*). RNAseq data confirmed partial knockdown of Atx2 mRNA in experimental flies expressing Atx2-RNAi (*Figure 3B*). Immunoblots might not reflect this at the protein level likely because the expression of RNAi is restricted to *elav* neurons, which are difficult to isolate and analyze, hence we verified Atx2 RNAi efficiency in the wing disc using *ptc-Gal4*. Atx2 levels were down only in the Gal4-expressing cells compared to the neighboring control cells (*Figure 3—figure supplement 1*).

Atx2 knockdown caused a significant reduction in levels of over 53.2% of the Atx2-target mRNAs, indicating a broad role for Atx2 in target-mRNA stabilization (*Figure 3C*). This was, however, not universal, and we observed that the levels of ~8.8% of the target mRNAs were increased with Atx2 knockdown. Non-target mRNA levels were substantially less affected: ~22.5% of all mRNAs from the global brain transcriptome were reduced by Atx2 knockdown, with ~57.2% showing no detectable change in expression levels (*Figure 3C*). In contrast, the analysis of nascent transcripts using intron reads showed that predominant Atx2 targets (~85%) remain unchanged (*Figure 3—figure supplement 3*), suggesting that the downregulation of target mRNAs in Atx2 RNAi is posttranscriptional.

To further validate the above results, TRIBE analysis was performed in *Drosophila* S2 cells. Atx2-ADARcd edited ~179 mRNA targets preferentially in the 3′UTR in S2 cells (*Figure 3—figure supplement 4A, B*). The targets identified from S2 cell TRIBE only marginally overlapped with the targets identified in the *Drosophila* brain (*Figure 3—figure supplement 4A*). This minimal overlap could be due to the RNA expression differences between the brain and S2 cells and/or target specificity of Atx2 in different cell types. It has been previously noticed that RNA-binding proteins can bind different RNA targets even within different neuronal populations (*McMahon et al., 2016*).

Like in brain, Atx2 silencing in S2 cells caused reduction in the levels of significant percentage of targets (~49%) compared to ~14% of total S2 transcripts (*Figure 3—figure supplement 5A, B*). In contrast, ~74% of targets that showed reduced expression in Atx2 RNAi were upregulated in Atx2 overexpression (*Figure 3—figure supplement 5B*).

Together, the RNAseq data (1) provide additional evidence in support of in vivo interactions between Atx2 and target mRNAs identified by TRIBE analysis and (2) are consistent with Atx2 binding to the ARE motif acting predominantly to stabilize target mRNAs.

ARE*Score* is a numerical assessment of ARE strength with high scores correlating to reduced RNA stability in reporter assays (*Spasic et al., 2012*). In support of a broad role for Atx2 in the regulation of ARE function in neurons, UTRs with high ARE*Score* are clearly enriched in the Atx2-target mRNAs compared to the general neural transcriptome (*Figure 3D*). Further, within the group of Atx2-target RNAs identified by TRIBE, higher ARE*Score* strongly predict mRNAs whose steady-state levels are reduced by Atx2 knockdown in brain and S2 cells (*Figure 3E, Figure 3—figure supplement 6*). These results indicate that Atx2 stabilizes a subset of its target mRNAs by binding to AREs.

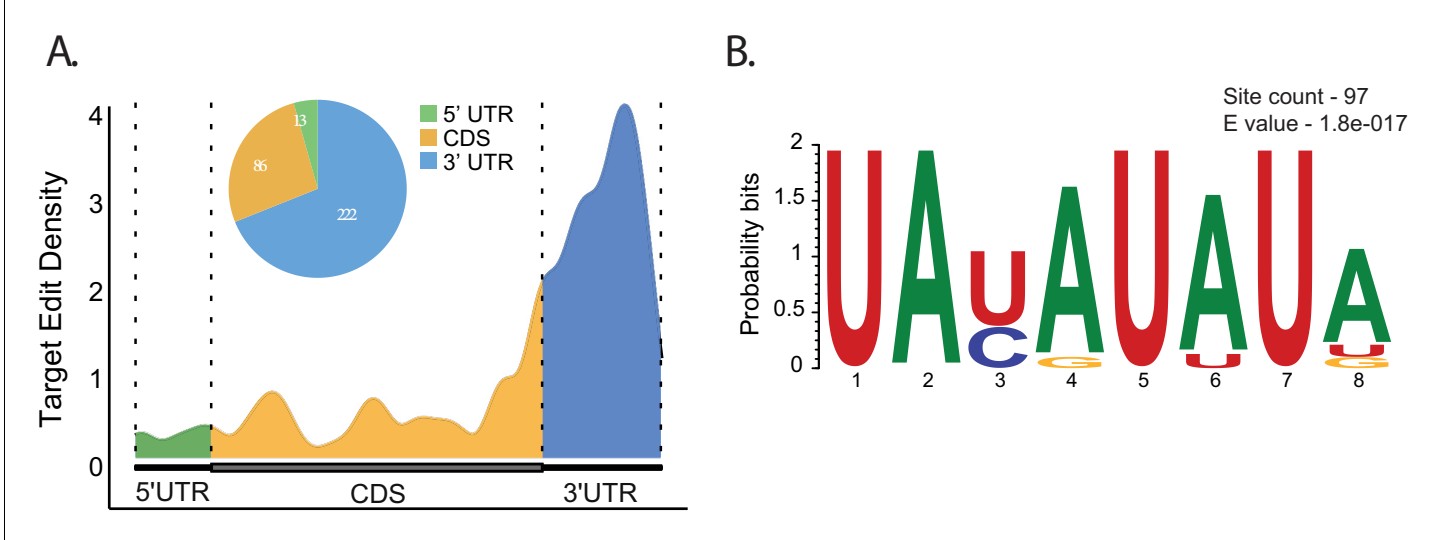

**Figure 2.** *Atx2* preferentially edits AU-rich sequences in the 3'UTRs of the target mRNAs. (**A**) Metaplot analysis showing Atx2 preferentially associates with 3'UTRs of the target mRNA. (**B**) Motif analyses involving ±100 bases around the edit site using MEME identifies AU-rich element sequences in the 3'UTRs of the target mRNAs.

The online version of this article includes the following figure supplement(s) for figure 2:

**Figure supplement 1.** Exon specificity of Atx2 mRNA interactions and sequence selectivity of observed Atx2–coding sequence (CDS) interactions.

**Figure supplement 2.** Atx2-target mRNAs have longer-than-average 3'UTRs.

**Figure supplement 3.** Coding sequence edits were co-existent with 3'UTR edits.

**Figure supplement 4.** Multiple-sequence alignment was performed for regions around Atx2 binding motifs in *Akh* and *Vmat* genes between *Drosophila melanogaster* and closely related *Drosophila* species.

The data also point to alternative and/or context-specific mechanisms for mRNA regulation by Atx2. For instance, some Atx2-target mRNAs lack AREs and a significant subset of ARE containing target mRNAs are not destabilized by Atx2 knockdown (*Figure 3—figure supplement 2*). These could be explained by alternative pathways by which Atx2 is recruited to mRNAs, for example, via microRNA pathway components (*McCann et al., 2011*; *Sudhakaran et al., 2014*) or by further layers of regulation conferred by additional RBPs recruited onto target mRNAs.

To further understand the mechanisms by which Atx2 interacts with its target mRNAs, we asked which domains of Atx2 might be required for this function. In particular, we asked whether the unstructured IDR or structured Lsm/LsmAD domains contributed to the specificity of Atx2–mRNA target interactions.

### The cIDR domain enables and the Lsm domain opposes Atx2 interactions with target mRNA

Atx2 has three structured domains (Lsm, LsmAD, and PAM2) embedded within extended, poorly structured regions of the protein (*Bakthavachalu et al., 2018*). Although interactions mediated by structured domains in vivo are necessary for normal organismal viability in *Drosophila*, the mechanism by which structure domains function is only clear for PAM2, which, by binding to polyA binding protein PABP, likely allows interactions with polyA tails of mRNAs (*Satterfield and Pallanck, 2006*). In contrast, the most prominent disordered regions in Atx2 (mIDR and cIDR) contribute little to animal viability but are selectively required for the assembly of mRNPs in neurons or cultured cells (*Bakthavachalu et al., 2018*). To test how Lsm, LsmAD, and cIDR domains of Atx2 contribute to the specificity of mRNA target interactions, we performed TRIBE analyses with specific domain-deleted forms of Atx2 (*Figure 4B*). Because purified Lsm domain alone as well as Lsm+LsmAD domains has been reported to bind the AU-rich sequences in vitro, we generated transgenes expressing a construct with Lsm+LsmAD domains of Atx2 fused to the catalytic domain of ADAR (*Yokoshi et al., 2014*). In addition, we created transgenic lines expressing Atx2 deleted for either Lsm, LsmAD, mIDR, or cIDR domains based on the UAS-Atx2-ADARcd construct scaffold (*Figure 4A*). Using the

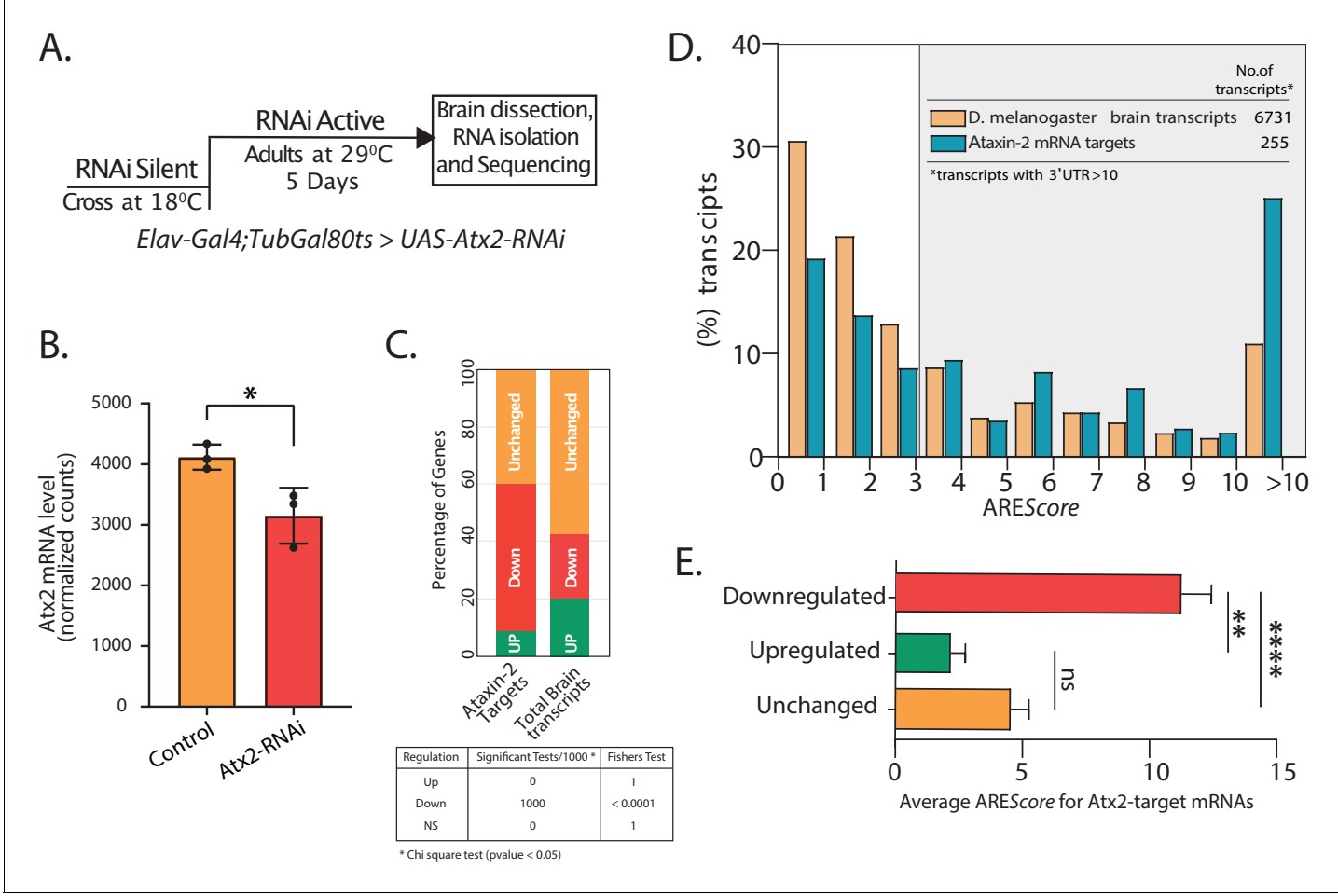

**Figure 3.** Atx2–*ARE* interactions modulate mRNA stability. (**A**) Schematic of strategy to induce RNAi expression specifically in adult fly brain. Total RNA extracted from brain post 5 days at 29°C was sequenced using Illumina 2500 and differential expression analyzed using DESeq2. (**B**) Normalized mRNA read counts showing Atx2 mRNA levels to be significantly reduced following RNAi expression compared to Gal4 control with a p-value of <0.0298 using Student's t-test. (**C**) Effect on Atx2-target mRNAs following Atx2 knockdown: the majority are reduced in level, indicating a role for Atx2 is target stabilization. Bootstraping was performed 1000 times with replacement. Statistics was performed using chi-square test with cutoff (p-value <0.05). Fisher's test was used to combine the p-values. (**D**) ARE*Score* analysis showing AU-rich elements (AREs) to be enriched in Atx2-target mRNAs compared to the brain transcriptome. (**E**) Among Atx2-target mRNAs, higher ARE scores are seen in mRNAs whose levels are reduced following Atx2 knockdown.

The online version of this article includes the following source data and figure supplement(s) for figure 3:

**Source data 1.** Data related to *Figure 3C*.
**Source data 2.** Data related to *Figure 3E*.
**Figure supplement 1.** *Atx2* RNAi knockdown reduces Atx2 expressions.
**Figure supplement 1—source data 1.** Data related to RNAseq mapping percentage from Gal4 and Atx2 RNAi brains.
**Figure supplement 2.** Correlation between ARE*Score* and relative expression levels of Atx2-downregulated targets for three biological replicates (selected based on a p-value<0.05).
**Figure supplement 3.** Differential expression analysis of intron reads from control and Atx2 RNAi show no major reduction in levels of Atx2 Targets of RNA-Binding Proteins Identified by Editing nascent mRNA targets.
**Figure supplement 3—source data 1.** Fold changes in the introns of Atx2-target mRNAs upon Atx2 RNAi.
**Figure supplement 4.** Comparative analyses of Atx2 targets in *Drosophila* brain and S2 cells.
**Figure supplement 4—source data 1.** Percentage edits of Atx2-target mRNAs in S2 cells among RNAseq replicates.
**Figure supplement 5.** Altering levels of Atx2 in S2 cells affects the levels of target mRNAs.
**Figure supplement 5—source data 1.** Fold changes in Atx2-target mRNAs when Atx2 is overexpressed or knocked down in S2 cells.
**Figure supplement 6.** ARE*Score* correlate directly with the stability of Atx2-target mRNAs in S2 cells.

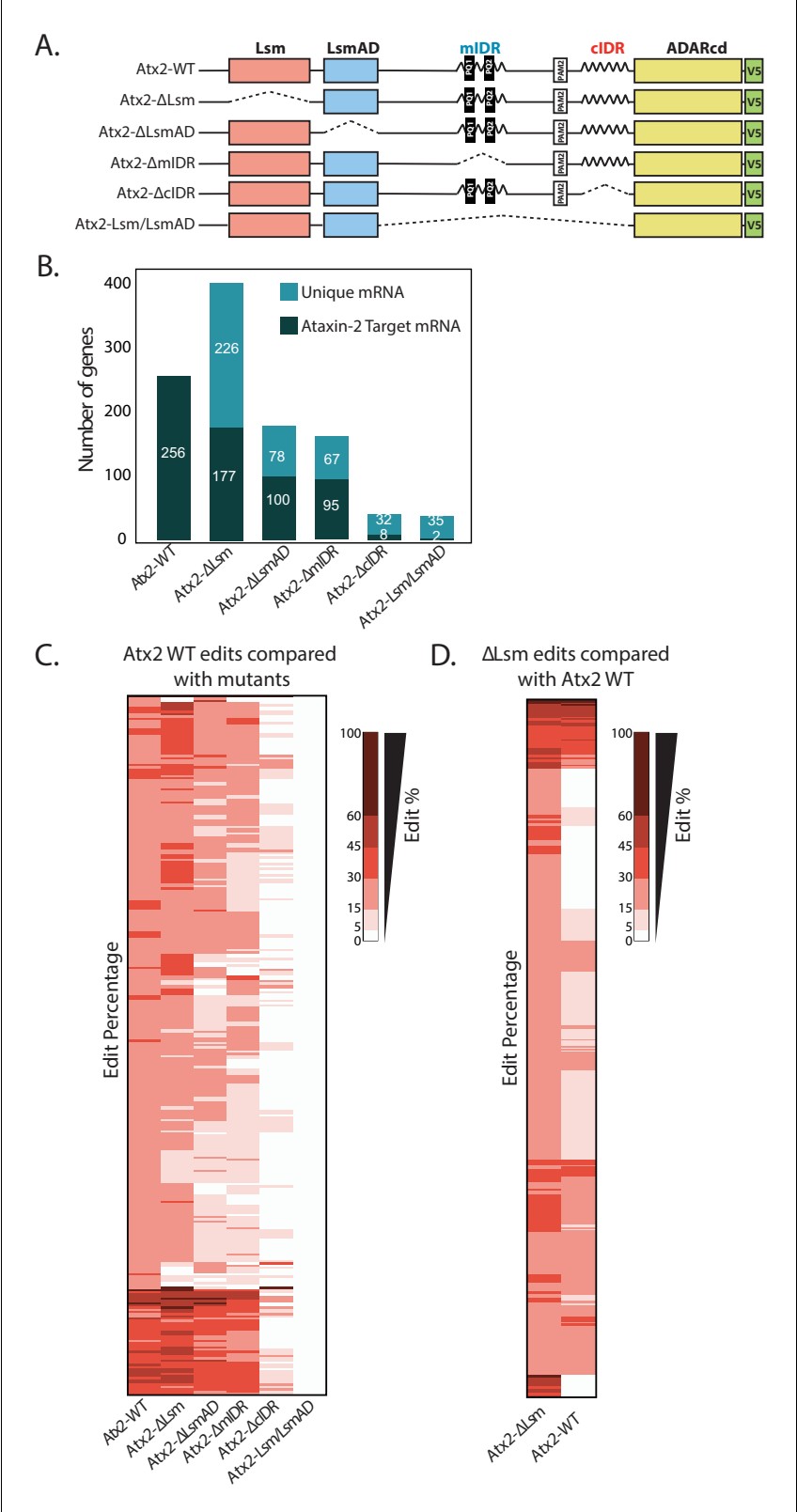

**Figure 4.** Atx2 requires its cIDR domain for interaction with its target mRNA. (**A**) An illustration of domains of Atx2. The structured (Like-Sm [Lsm] and Lsm-associated [LsmAD]) and disordered domains (mIDR and cIDR) of Atx2 protein were deleted one at a time to understand the domains necessary for its interaction with target RNA. The wild-type Atx2 and the deletions contained a c-terminal ADARcd and V5 tag. The deletions are shown by the dotted lines. (**B**) Deletion of LsmAD or mIDR domain reduced the ability of Atx2 to interact with its targets, while cIDR deletion almost entirely

*Figure 4 continued on next page*

Figure 4 continued

prevented Atx2 mRNA interactions. The Lsm-domain deletion showed an overall increase in mRNA edits. (C) More detailed heatmap view of individual target mRNAs showing target edit percentages in flies expressing control or domain-deleted forms Atx2 fused to ADARcd. (D) Most of the apparently new targets identified in Atx2ΔLsm Targets of RNA-Binding Proteins Identified by Editing analyses are also edited, albeit at a lower efficiency, in flies expressing full-length Atx2-ADAR fusions. This suggests that deletion of the Lsm domain increases the interaction of Atx2 with its native target mRNAs. (For C and D, 15% threshold edits identified for samples in column 1 [Atx2 WT in C and Atx2ΔLsm in D] was used to compare with all the edits of the rest of the columns to generate a heatmap.)

The online version of this article includes the following source data and figure supplement(s) for figure 4:

**Source data 1.** Data related to *Figure 4B*.
**Source data 2.** Data related to *Figure 4C, D*.
**Figure supplement 1.** Atx2 Like-Sm (Lsm), Lsm-associated (LsmAD), or mIDR domains are not essential for AU-rich motif preference.
**Figure supplement 1—source data 1.** Mapping percentage for different Atx2 deletions.
**Figure supplement 2.** Atx2ΔLsm unique targets were present at a reduced edit threshold in Atx2 WT.
**Figure supplement 3.** Equivalent read quality and ADAR expression across Atx2 deletion analyses.

same approach as earlier, we analyzed how each of the domain deletions affected the Atx2-target binding in adult neurons.

Initial observations indicated that Lsm+LsmAD domains on their own could not efficiently target ADAR to mRNA: transcripts sequenced from Lsm+LsmAD-ADARcd-expressing brains contained negligible edits, which did not overlap significantly with Atx2 targets (*Figure 4B*). Therefore, the Lsm and LsmAD domains are insufficient to drive Atx2–mRNA interactions in vivo. Further, deletions of either Lsm or LsmAD did not block the ability of Atx2 to interact with most TRIBE targets: thus, they appear neither necessary nor sufficient for Atx2 targeting to these mRNAs (*Figure 4B, C*). These surprising observations led us to examine the role of disordered domains in driving Atx2–mRNA interactions in vivo.

In contrast to the effects of deleting the Lsm domain, deletion of cIDR abolished Atx2 binding to most target mRNAs (*Figure 4B, C*). As the cIDR plays a major role in mRNP assembly (*Bakthavachalu et al., 2018*), this unexpected observation suggests that Atx2 moves into proximity of target mRNAs only after cIDR-mediated granule formation. Deletion of the mIDR resulted only in a relatively minor reduction in target binding, but this is consistent with prior work indicating only a minor role in mRNP assembly (*Bakthavachalu et al., 2018*). In addition, the deletion of Lsm, LsmAD, or mIDR domains did not alter AU-rich motif preference for Atx2, suggesting that these domains do not provide mRNA binding specificity (*Figure 4—figure supplement 1*). Deeper analyses provided additional support for a model in which Atx2 associates to RNA-binding proteins in individual mRNPs, but is brought into closer contact with mRNAs through remodeling events associated with the formation of higher order mRNP assemblies.

A key observation is that while Lsm-domain deletions did not reduce RNA edits, they also curiously resulted in a significantly larger number of edited target mRNAs compared to the full-length, wild-type Atx2 (*Figure 4C*). A more detailed analysis showed that several of the apparently novel targets of Atx2ΔLsm were also bound by the wild-type Atx2 but with reduced affinity and were therefore below the threshold of our analysis (*Figure 4D, Figure 4—figure supplement 2*). Because sequencing depth, read quality, and ADAR mRNA and protein levels were similar across control and domain-deletion experiments (*Figure 4—figure supplement 3A–C*), these observations argue that the Lsm domain acts to broadly antagonize physiologically relevant mRNA interactions driven by the cIDR. Thus, while the cIDR domain of Atx2 is essential for its mRNA target interactions, the Lsm domain is inhibitory: in its absence, Atx2 shows an enhanced association with its native, target mRNAs.

## The Atx2 Lsm domain inhibits cIDR-mediated mRNP granule assembly

The most parsimonious explanation for the observed opposing effects of Lsm and cIDR domain deletions on mRNA editing (*Figure 4*) is that the Lsm domain functions to oppose cIDR-mediated mRNP granule assembly (*Bakthavachalu et al., 2018*). To directly test this hypothesis, we first asked whether deletion of the Lsm domain enhanced Atx2 RNP assembly. Expression of wild-type C-terminally SNAP tagged Atx2 (Atx2-SNAP) in S2 cells led to the formation of SG-like RNP-granule foci through a mechanism dependent on cIDR as described previously (*Figure 5ii, v*;

*Bakthavachalu et al., 2018*). Strikingly consistent with our predictions, expression of Atx2ΔLsm-SNAP constructs lacking the Lsm domain induced significantly larger RNP granules compared to wild-type Atx2-SNAP (*Figure 5i*). Similar to wild-type Atx2-SNAP granules, these large Atx2ΔLsm-SNAP-induced granules also contained the SG protein, G3BP/Rasputin (Rin) (*Figure 5—figure supplement 1*), and their formation required the presence the cIDR (*Figure 5iv*). Thus, Lsm domain appears to act antagonistically to cIDR to prevent RNP granule assembly.

To further confirm this, we tested if the inclusion of additional Lsm domains would reduce the ability of Atx2 to form RNP granules. As the deletion of the Atx2 mIDR does not alter mRNP granule assembly in S2 cells (*Bakthavachalu et al., 2018*), we replaced mIDR with an additional Lsm domain to create an Atx2 protein with two Lsm domains. Remarkably, Atx2 with two Lsm showed much smaller foci in S2 cells (*Figure 5Aiii*). Quantification of granules in these cells reiterated that while ΔLsm formed fewer but larger granules, increasing the valency of Lsm domains led to the formation

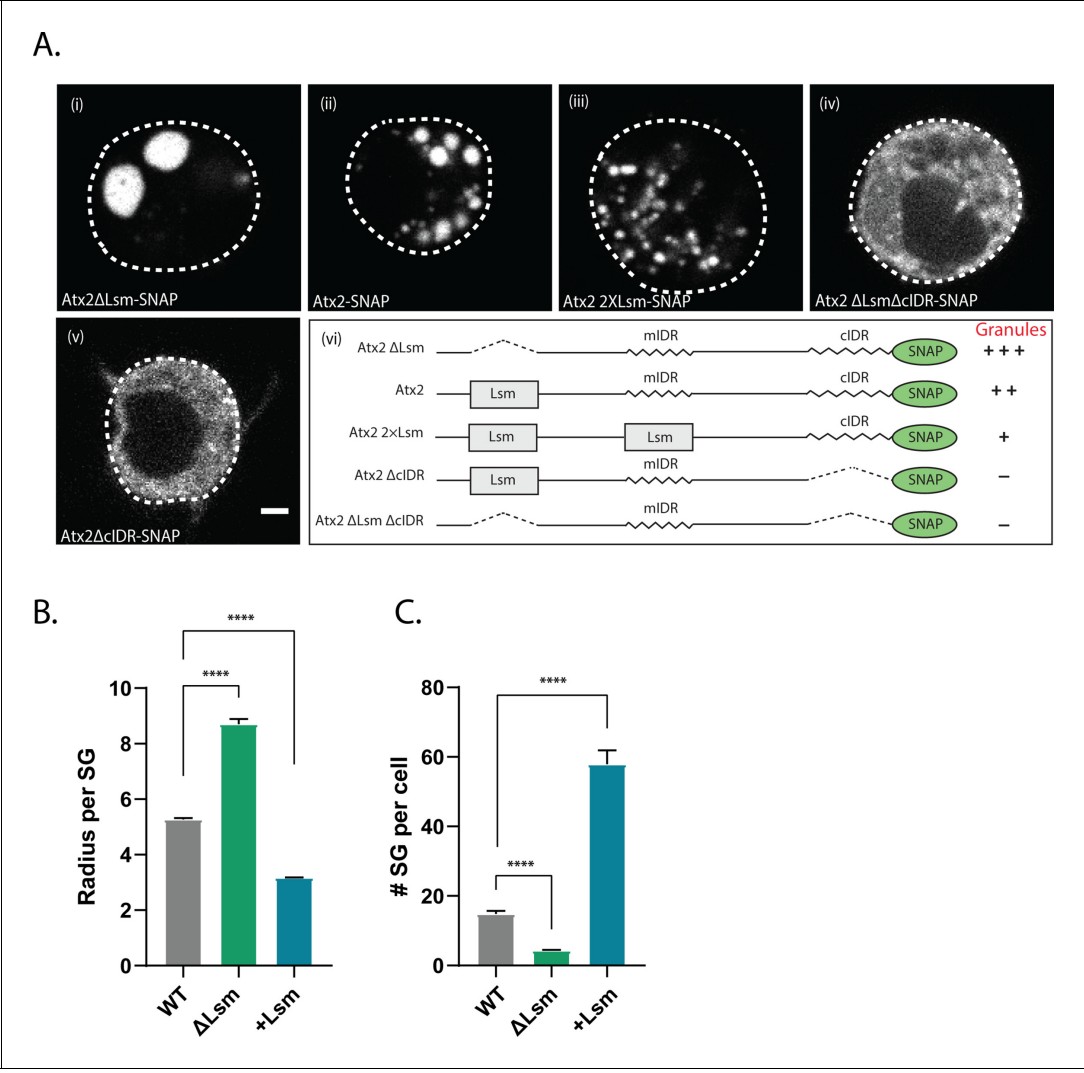

**Figure 5.** Atx2 Like-Sm (Lsm) domain alters granule dynamics. (**A**) *Drosophila* S2 cells expressing Atx2 protein with a C-terminal SNAP tag. Atx2ΔLsm form large cytoplasmic granules (i) that are much larger than WT Atx2 granules (ii). Atx2 with an additional Lsm domain in place of mIDR to create 2XLsm forms smaller granules compared to WT Atx2 in S2 cells (iii). Deletion of cIDR domain blocks Atx2 granule formation (iv). In the absence of cIDR, Lsm deletion does not rescue Atx2 granules (v). Domain map along with the granule phenotype is shown in (vi). The scale bar corresponds to 2 μm. (**B**) Radius of the stress granule (size) and (**C**) number of stress granules per cell were quantified and plotted. More than 80 cells per genotype were used for granule quantification.

The online version of this article includes the following figure supplement(s) for figure 5:

**Figure supplement 1.** Granules formed by overexpression of Atx2 and ΔLsm resemble stress granules (SGs) that also contain SG protein Rasputin (Rin).

of several smaller granules (*Figure 5B*). The summary of these results is shown in *Figure 5Avi*. Taken together, the observations that deletion of the Lsm enhances and addition of an extra Lsm domain inhibits RNP assembly provide strong support for two opposing activities encoded by the Lsm and cIDR domains of Atx2.

## Discussion

Previous work has shown that *Drosophila* Atx2 functions in neurons as a translational activator of the *period* mRNA that controls circadian rhythms and as a translational repressor of the calcium-calmodulin-dependent kinase CaMKII involved in synaptic plasticity and memory (*Lim and Allada, 2013*; *Sudhakaran et al., 2014*; *Zhang et al., 2013*). Atx2 is also required for the assembly of neuronal mRNPs believed to provide pools of synaptically localized mRNAs whose translation contributes the consolidation of long-term memory (*McCann et al., 2011*). These studies indicate the specific positive and negative translational functions of Atx2, which are mediated by structured–domain interactions with Lsm12 or Me31B/DDX6, respectively (*Lee et al., 2017*), and that mRNP-assembly functions are mediated by its cIDR (*Bakthavachalu et al., 2018*). However, the generality of these mechanisms, the range of neuronal mRNAs and neuronal functions under Atx2 regulation, as well as how structured and disordered domain interactions are coordinated remain largely unknown. Here, by deploying and building on TRIBE analysis to identify a suite of Atx2-target mRNAs, we address these questions and provide insights of relevance for biology, technology, and medicine.

### Neural functions of Ataxin-2

TRIBE allows in vivo RNA targets of RBPs to be identified from small tissue samples, eliminating several technical challenges and artifacts associated with immunoprecipitation-based methods (*Jin et al., 2020*; *McMahon et al., 2016*; *Xu et al., 2018*). This method led to the identification of 256 *Drosophila* brain mRNAs that associate with Atx2 with the proximity and stability required for Atx2-linked enzymatic editing of the mRNA. These mRNAs are reproducibly identified in replicate experiments and do not show any over-representation of highly expressed mRNAs. Moreover, the observation that a substantial fraction of these mRNAs either have AREs in their 3'UTRs and/or show altered steady-state levels following Atx2 knockdown argues that the majority represent real Atx2 targets and not non-specific proximity-based editing events that can sometimes occur within RNP complexes (*Biswas et al., 2020*). Thus, the resulting robust data set of Atx2 targets may provide valuable hypotheses for biological functions and genetic pathways regulated by Atx2. For instance, a striking enhancement of mRNAs encoding specific neuropeptides and neuronal hormones suggests that altered intercellular communication mediated by their translational regulation may contribute to the behavioral plasticity associated with circadian time or long-term memory. Similarly, a large subset of target mRNAs encoding proteins regulating neural excitability through multiple different mechanisms is unexpected and points to the possibility that activity-regulated translation may act via local changes in membrane properties to achieve localized plasticity required for encoding specific memories. It is important to note that this analysis may miss mRNAs that are strong targets in a small subset of neurons but not in others, for instance, the other cells may express RBPs that prevent Atx2 interactions. Thus, more targets may be found by new approaches using TRIBE for single-cell analyses (*McMahon et al., 2016*).

But not all Atx2-regulated mRNAs have been identified. It is notable that two of the best-established Atx2 targets, CaMKII and *per*, were not identified by TRIBE. While the *per*–Atx2 interactions, being time- and cell-type restricted (*Lim and Allada, 2013*; *Zhang et al., 2013*), could potentially be missed for statistical reasons, this is unlikely the case for CaMKII, a highly expressed mRNA that co-immunoprecipitates with Atx2 (*Sudhakaran et al., 2014*). We suggest instead that these represent Atx2 targets missed by TRIBE because they are regulated through relatively indirect mechanisms that do not require close contact between Atx2 and the mRNA. For instance, in case of *per*, its 3'UTR is recognized by the sequence-specific RBP Twentyfour (TYF), which recruits Atx2 that in turns recruits a Lsm12-containing complex to the *per* 3'UTR, thus allowing translational activation (*Lee et al., 2017*). Similarly, for CaMKII, Atx2 may be recruited by miRNA pathway components and act via co-regulators such as Me31B/DDX6, through mechanisms that do not rely on close proximity between Atx2 and target mRNAs. The above may also help explain why previously proposed target mRNAs in metabolic pathways for instance are not represented in this data set (*Yokoshi et al.,*

*2014*). Indeed, Atx2 likely binds to several additional neuronal mRNAs not identified by TRIBE, which requires Atx2 proximity to the mRNA. Such targets may be better identified by CLIP-based methods. However, considerable new understanding can be provided by the detailed analysis of the 256 robust targets identified here by TRIBE.

One important insight is the discovery of a broad function for Atx2 in neuronal mRNA stabilization. Atx2 associates preferentially to 3′UTRs of the target-mRNAs, and particularly to AU-rich sequences (AREs) in these UTRs (*Figure 2*). AREs are common *cis*-regulatory features regulating mRNA stability, a posttranscriptional gene regulation strategy adopted by all eukaryotic cells (*García-Mauriño et al., 2017*). The observation that knockdown of Atx2 in *Drosophila* brain and S2 cells causes levels of a large fraction of the Atx2-target mRNAs to be significantly reduced (*Figure 3B, C*, *Figure 3—figure supplement 5B*) and that the most downregulated targets have strong ARE*Score* (*Spasic et al., 2012*) suggests that Atx2 directly or indirectly associates with AREs to protect mRNAs from degradation (*Figure 3E*). This could be achieved by blocking ARE-dependent recruitment of RNA degradation complexes through a mechanism similar to that described previously for HuR (*Peng et al., 1998*). These conclusions may also be relevant for mammalian Atxn2 as physical interactions between mammalian Atxn2 and AREs have been described previously using PAR-CLIP analyses from cultured HEK293 cells (*Yokoshi et al., 2014*). Moreover, Atxn2-CAG100-KnockIn mouse engineered to express polyQ expanded forms of Atxn2 that should enhance granule formation show a predominant upregulation of mRNAs, consistent with a role for Atx2-mediated mRNP assembly in stabilizing target mRNAs (*Sen et al., 2019*).

It is important to note that some mRNAs with high ARE scores do not appear to be stabilized by Atx2, and conversely, some that do not contain AREs appear to be affected by Atx2 knockdown (*Figure 3E*). Both of these observations are consistent with additional layers and mechanisms of regulation conferred by co-regulating RBPs: either by providing an alternative pathway for ARE regulation via, for instance, miRNA binding (*Sun et al., 2010*; *van Kouwenhove et al., 2011*) or an alternative mechanism for recruitment of Atx2 to the 3′UTR of mRNAs.

## Mechanisms of Ataxin-2 function in RNP granule assembly

Our work provides two insights into the mechanisms of mRNP formation. First, it indicates that individual mRNPs may be substantially remodeled as they assemble into higher order mRNP assemblies. In support of this, we show that Atx2 lacking its cIDR, which cannot form granules, is also not associated closely enough with mRNAs to allow their editing by a linked ADAR catalytic domain. One possibility is that mRNP remodeling is driven by major conformational changes in RBPs, which not only increase their propensity to drive mRNP condensation but also result in altered RBP–RBP and RBP–RNA interactions. In this context, recent work on G3BP/Rin has shown that the protein exists in two dramatically different conformational states: a closed form, in which its IDRs are inaccessible for condensation reactions, and a dephosphorylation-induced open form, capable of mediating SG association (*Guillén-Boixet et al., 2020*; *Laver et al., 2020*; *Sanders et al., 2020*). In such a framework, it is easy to see how Atx2 interactions with RBPs and mRNAs could be altered under conditions that support granule assembly. While these changes in Atx2 interactions could occur due to structural changes in other components of Atx2-containing mRNPs, our second insight is that alterations in the Atx2 protein itself probably occur and contribute to driving granule assembly.

Ataxin-2 is a modular protein capable of association with multiple translational control components (*Dansithong et al., 2015*; *Lastres-Becker et al., 2016*; *Lee et al., 2017*; *Satterfield and Pallanck, 2006*; *Swisher and Parker, 2010*). Although Atx2 lacks RNA recognition domains like RRMs, KH, or other previously characterized RNA-binding domains, homology-based modeling studies and indirect experimental observations have suggested that the Lsm domain of Atx2 may mediate RNA interaction (*Calabretta and Richard, 2015*; *Hentze et al., 2018*; *Yokoshi et al., 2014*). However, direct experimental tests of this hypothesis show that close Atx2 interactions with mRNA, as assessed by TRIBE, are actively prevented by the Lsm domain, which also opposes mRNP assembly (*Figure 4* and *Figure 5*). In contrast, the cIDR that drives mRNP assembly is necessary for Atx2-coupled editing of target mRNA. An untested prediction of this model is that the TRIBE analysis of Atx2 forms carrying Lsm domain repeats would yield results similar to those seen after cIDR deletion.

The simplest explanation for these findings is that Atx2 association with individual, potentially translationally active mRNPs in the soluble phases is mediated by Lsm domain–RBP interactions that also occlude or prevent cIDR-mediated mRNP assembly (*Ciosk et al., 2004*; *Lee et al., 2017*;

*Satterfield and Pallanck, 2006*). Conditions that promote mRNP assembly disrupt Lsm-domain-mediated interactions and enable cIDR-driven granule formation (*Figure 6*). We note that recent work on G3BP has beautifully elaborated phosphorylation-regulated intramolecular interactions that similarly allow the molecule to switch between soluble and assembly-competent conformations (*Guillén-Boixet et al., 2020*; *Laver et al., 2020*; *Sanders et al., 2020*). Though our experiments do not yet define molecular and biophysical details by which Atx2 transitions from assembly-inhibited to assembly-competent states, our observations (1) clearly demonstrate crucial opposing, physiological roles of the Lsm and cIDR domains in this process and (2) suggest that regulation of intermolecular interactions mediated by the Lsm domain will be involved in control of Atx2-mediated granule assembly.

It is important to note that Ataxin-2 has additional essential functions beyond those described here. In particular, given that the Atx2 structured domains not required for TRIBE-target binding are essential for survival, unlike the IDR, which is required for editing of TRIBE targets but not for animal survival, it appears likely that a class of Atx2-target mRNAs is regulated outside of mRNP granules through largely structured-domain interactions (*Figure 6*). Additional approaches and experiments are required to identify such mRNAs as well as mechanisms by which they are regulated.

## Insight for disease and therapeutics

Ataxin-2 has attracted considerable clinical interest for three main reasons. First, assembly-promoting mutations in the Ataxin-2 gene or associated RNA binding and SG proteins such as TDP-43 can cause neurodegenerative disease. Second, SG proteins such as TDP-43 are usually present in cytoplasmic protein inclusions associated with familial and heritable forms of ALS and FTD. Third, reduction of Ataxin-2 can slow neurodegeneration in animal models of ALS, indicating that the normal

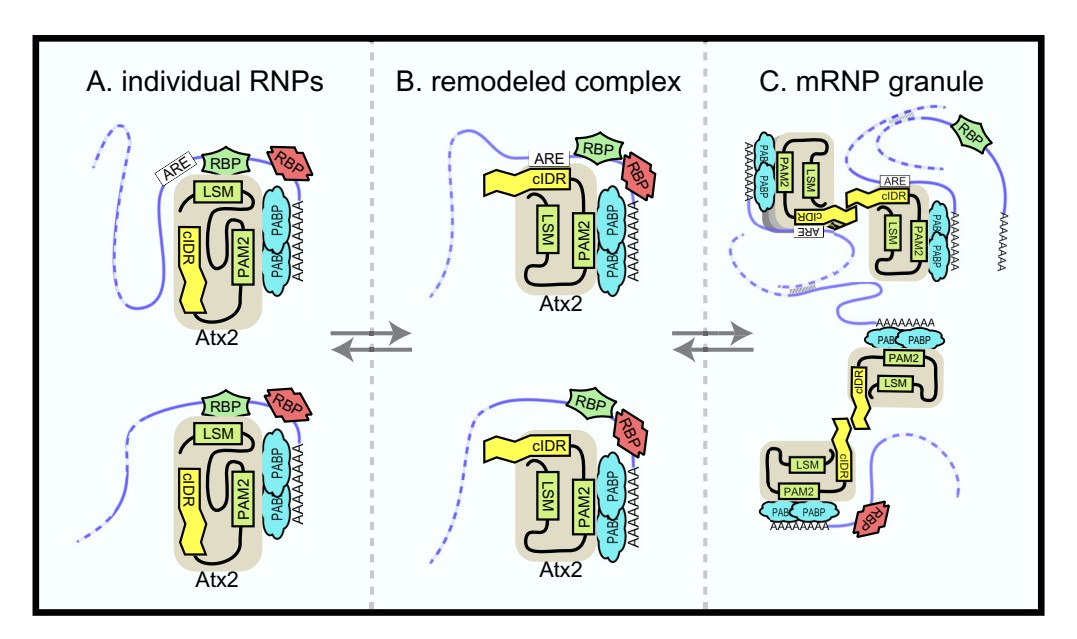

**Figure 6.** Ataxin-2 domains in mRNA target regulation. A model to explain the current and previous observations on Ataxin-2 as well as its function in mRNA regulation. (**A**) Ataxin-2 does not make direct contact with mRNAs in soluble mRNPs. Instead, it is recruited to RNA by other RBPs that bind to structured Like-Sm (Lsm) (or Lsm-associated [LsmAD]) domains. In these soluble mRNPs, the cIDR is buried and inaccessible. One class of mRNP (above) contains AU-rich elements (AREs); the second class below (e.g., *per* in *Drosophila*) does not. (**B**) Under specific signaling conditions (e.g., stress), RBP–Atx2 interactions are prevented. In these 'remodeled' mRNPs, the Atx2-cIDR is exposed. We speculate that a segment of this intrinsically disordered domain (IDR) directly binds to nearby AREs (or to ARE-binding proteins), while other segments of the IDR mediate multivalent interactions that contribute to mRNP condensation. (**C**) Ataxin-2 cIDR interactions enable mRNP assembly into granules, facilitated by RNA–RNA crosslinks and interactions mediated by other RBPs (e.g., G3BP/Rin). These RNP granules may include both ARE-containing mRNAs (edited) and mRNAs that do not. Note that additional RBPs (red) could determine not only Atx2 proximity to specific mRNAs (leading to editing) but also the effect of Atx2 on mRNA translation and/or stability. These and alternative models remain to be tested.

function of Ataxin-2 is required for initiation or progression of disease (*Becker et al., 2017*; *Scoles et al., 2017*). These findings, interpreted in a framework wherein SGs are thought to facilitate the nucleation of pathogenic amyloid filaments, have led to the development of therapeutics based on reducing levels of Ataxin-2, for example, using antisense oligonucleotides by major companies such as Ionis and Biogen Inc. In this context, the discovery that the Lsm domain inhibits mRNP assembly suggests, first, that mutations inactivating this domain could have effects similar to polyQ expansions and promote disease and, second, that compounds targeting specific domains and activities of Ataxin-2 may prove more effective as therapeutics than those that knock down protein levels.

The case for our understanding of the function of each Atx2 domain and developing specific modulators is particularly strong since Ataxin-2 protein itself has several important roles not only in mRNA stabilization, as shown here, but also in protein translation, cell signaling, metabolism, and embryonic development (*Halbach et al., 2017*; *Inagaki et al., 2020*; *Kato et al., 2019*; *Lim and Allada, 2013*; *Meierhofer et al., 2016*; *Yang et al., 2019*; *Yokoshi et al., 2014*; *Zhang et al., 2013*), as shown by various biological studies of native Ataxin-2 function.

# Materials and methods

## Key resources table

| Reagent type (species) or resource | Designation | Source or reference | Identifiers | Additional information |
|---|---|---|---|---|
| Genetic reagent (*Drosophila melanogaster*) | *UAS-Atx2-WT-ADARcd* | This paper | N/A | Related to *Figures 1*, *2* and *4* |
| Genetic reagent (*Drosophila melanogaster*) | *UAS-Atx2-ΔLsm-ADARcd* | This paper | N/A | Related to *Figure 4* |
| Genetic reagent (*Drosophila melanogaster*) | *UAS-Atx2-ΔLsm-AD-ADARcd* | This paper | N/A | Related to *Figure 4* |
| Genetic reagent (*Drosophila melanogaster*) | (*UAS-Atx2-ΔmIDR-ADARcd*) | This paper | N/A | Related to *Figure 4* |
| Genetic reagent (*Drosophila melanogaster*) | *UAS-Atx2-ΔcIDR-ADARcd* | This paper | N/A | Related to *Figure 4* |
| Genetic reagent (*Drosophila melanogaster*) | *UAS-Atx2-only Lsm/Lsm-AD -ADARcd* | This paper | N/A | Related to *Figure 4* |
| Genetic reagent (*Drosophila melanogaster*) | *Elav-Gal4; Tub-Gal80^{ts}* | Bloomington *Drosophila* Stock center | | Related to *Figure 3* |
| Genetic reagent (*Drosophila melanogaster*) | *UAS-Atx2 RNAi* | Vienna *Drosophila* RNAi Center stock collection | 34955 | Related to *Figure 3* |
| Cell line (*Drosophila melanogaster*) | S2R+ cells | DGRC | RRID:CVCL_Z831 | |
| Recombinant DNA reagent | pJFRC7-20XUAS-IVS-8_Atx2-ADARcd (plasmid) | This paper | N/A | Construct to express WT Atx2-ADAR fusion protein |
| Recombinant DNA reagent | pJFRC7-20XUAS-IVS-8_Atx2ΔmIDR -ADARcd (plasmid) | This paper | N/A | Construct to express Atx2ΔmIDR-ADAR fusion protein |

*Continued on next page*

Continued

| Reagent type (species) or resource | Designation | Source or reference | Identifiers | Additional information |
|---|---|---|---|---|
| Recombinant DNA reagent | pJFRC7-20XUAS-IVS-8_Atx2ΔcIDR -ADARcd (plasmid) | This paper | N/A | Construct to express Atx2ΔcIDR-ADAR fusion protein |
| Recombinant DNA reagent | pJFRC7-20XUAS-IVS-8_Atx2ΔLsm -ADARcd (plasmid) | This paper | N/A | Construct to express Atx2ΔLsm-ADAR fusion protein |
| Recombinant DNA reagent | pJFRC7-20XUAS-IVS-8_Atx2ΔLsmAD-ADARcd (plasmid) | This paper | N/A | Construct to express Atx2ΔLsmAD-ADAR fusion protein |
| Recombinant DNA reagent | pJFRC7-20XUAS-IVS-8_Atx2only-Lsm-LsmAD-ADARcd (plasmid) | This paper | N/A | Construct to express only Lsm-LsmAD-ADAR fusion protein |
| Antibody | Anti-Atx2 (chicken polyclonal) | *Bakthavachalu et al., 2018* | | IF (1:500) WB (1:1000) |
| Antibody | Anti-Rasputin (rabbit polyclonal) | *Aguilera-Gomez et al., 2017* | | IF (1:500) |
| Antibody | Anti-GFP (rabbit polyclonal) | Molecular probes | Cat# A11122 | IF (1:500) |
| Antibody | Anti-GFP (chicken polyclonal) | Abcam | Cat# mAb 13970 | IF (1:500) |
| Antibody | Anti-V5 (rabbit polyclonal) | Santa Cruz Biotechnology | Cat# sc83849-R | IF (1:500) WB (1:1000) |
| Antibody | Anti-nc82 (mouse monoclonal) | *Kittel, 2006* | | IF (1:500) |
| Antibody | Anti-tubulin (mouse monoclonal) | DSHB | Cat# E7C | WB (1:2000) |
| Antibody | Alexa Fluor 555 (polyclonal goat anti-chicken IgG) | Invitrogen | Cat# A21437 | IF (1:1000) |
| Antibody | Alexa Fluor 488 (polyclonal goat anti-chicken IgG) | Invitrogen | Cat# A11039 | IF (1:1000) |
| Antibody | Alexa Fluor 647 (polyclonal goat anti-chicken IgG) | Invitrogen | Cat# A21449 | IF (1:1000) |
| Antibody | Alexa Fluor 555 (polyclonal goat anti-rabbit IgG) | Invitrogen | Cat# A21428 | IF (1:1000) |
| Antibody | Alexa Fluor 488 (polyclonal goat anti-rabbit IgG) | Invitrogen | Cat# A11078 | IF (1:1000) |
| Antibody | Alexa Fluor 647 (polyclonal goat anti-rabbit IgG) | Invitrogen | Cat# A21244 | IF (1:1000) |
| Antibody | Alexa Fluor 555 (polyclonal goat anti-mouse IgG) | Invitrogen | Cat# A21422 | IF (1:1000) |
| Antibody | Alexa Fluor 488 (polyclonal goat anti-mouse IgG) | Invitrogen | Cat# A21121 | IF (1:1000) |

*Continued*

| Reagent type (species) or resource | Designation | Source or reference | Identifiers | Additional information |
|---|---|---|---|---|
| Antibody | Alexa Fluor 647 (polyclonal goat anti-mouse IgG) | Invitrogen | Cat# A21235 | IF (1:1000) |
| Chemical compound | Vectashield Mounting Medium | Vector Laboratories | Cat# H-1000 | |
| Chemical compound | SNAP-Surface 549 | New England Biolabs | Cat# S9112S | IF (1:500) |
| Software, algorithm | TRIBE | *McMahon et al., 2016* | https://github.com/rosbashlab/TRIBE | |
| Software, algorithm | STAR v2.5.3 | *Dobin et al., 2013* | https://github.com/alexdobin/STAR | |
| Software, algorithm | HTSeq v0.11.2 | *Anders et al., 2015* | https://github.com/htseq/htseq | |
| Software, algorithm | DESeq2 | *Love et al., 2014* | https://bioconductor.org/packages/release/bioc/html/DESeq2.html | |
| Software, algorithm | AREScore | *Spasic et al., 2012* | http://arescore.dkfz.de/arescore.pl | |
| Software, algorithm | Guitar | *Cui et al., 2016* | https://bioconductor.org/packages/release/bioc/html/Guitar.html | |
| Software, algorithm | Bedtools | *Quinlan and Hall, 2010* | https://github.com/arq5x/bedtools2 | |
| Software, algorithm | twoBitToFa | - | https://genome.ucsc.edu/goldenPath/help/twoBit.html | |
| Software, algorithm | MEME suite | *Bailey et al., 2009* | http://meme-suite.org/tools/meme | |
| Software, algorithm | Cellprofiler | *McQuin et al., 2018* | https://cellprofiler.org | |
| Software, algorithm | ImageJ | *Schneider et al., 2012* | https://imagej.nih.gov/ij/ | |
| Software, algorithm | Ggplot2 | *Wilkinson, 2011* | https://github.com/tidyverse/ggplot2 | |
| Software, algorithm | Pheatmap | | https://cran.r-project.org/web/packages/pheatmap/index.html | |
| Software, algorithm | SnapDragon | | https://www.flyrnai.org/snapdragon | |

## Generation and rearing of *Drosophila* stocks

*Drosophila* stocks were maintained at 25°C in corn meal agar, and experimental fly crosses were done as specified in the respective experimental methods. The list of *Drosophila* stocks used and transgenic flies generated for this study are given in Key resources table.

## S2 cell culture

*Drosophila* S2R+ cells were obtained from DGRC and cultured in Schneider's medium with 10% FBS, penicillin, and streptomycin at 25°C.

## Creation of transgenic animals

*Drosophila* Atx2 full-length cDNA was cloned into pJFRC7-20XUAS-IVS-8_ADARcd plasmid (a gift from Rosbash Lab) to create pJFRC7-20XUAS-IVS-8_Atx2wt-ADARcd plasmid. Domain deletions

were created using overlapping PCR and Gibson assembly or non-overlapping PCR and ligation using pJFRC7-20XUAS-IVS-8_Atx2wt-ADARcd as template. Sequence-confirmed plasmids were used to generate transgenic *Drosophila* using PhiC31 integrase-dependent site-specific insertion of the transgene on the second chromosome. Details of plasmids used for transgenesis are listed in Key resources table. Embryo injections were performed at NCBS transgenic fly facility. Primers used for domain deletions are listed in Key resources table. The sequences of primers used for generating Atx2 domain deletions are provided in *Supplementary file 1*.

## Experimental fly crosses

Strains homozygous for the *elav-Gal4* and *tub-Gal80^{ts}* transgenes were crossed with homozygous UAS-transgenic flies at 18°C till the adult fly emerged. One-day-old adult flies from the crosses were maintained at 29°C for 5 days before processing for RNA extraction.

## S2 cell transfections for immunofluorescence

Half a million cells were transfected with 500 ng plasmid using Mirus TransIT-X2 Dynamic Delivery System (MIR6000) as per the manufacturer's instructions. The cells were harvested 24 hr after transfection and processed for immunofluorescence.

## Double-stranded (ds) RNA generation and S2 cell transfection

Mock and Atx2 RNAi was performed using dsRNA produced by in vitro transcription (IVT). For mock, we used a GFP sequence from open reading frame. Atx2 RNAi target sites were chosen using SnapDragon tool (https://fgr.hms.harvard.edu/snapdragon). PCR-generated DNA template containing the T7 promoter sequence at both the ends was used as IVT template for dsRNA synthesis using Megascript T7 High Yield Transcription kit (AM1334; Invitrogen). Half a million cells were transfected with 5 µg of mock or *Atx2* dsRNA using Effectene Transfection reagent (Qiagen 301425) as per the manufacturer's instructions. After 48 hr of the first round of transfections, cells were again transfected with 5 µg of respective dsRNAs. The sequences of primers used for generating dsRNAs are provided in *Supplementary file 1*.

## RNA extraction from brain and NGS

Total RNA was isolated from adult brain (10–12 brains per replicate) dissected in RNA Later using TRIzol reagent (Invitrogen) as per the manufacturer's protocol. Illumina libraries were prepared from Poly(A)-enriched mRNA using NEBNext Ultra II Directional RNA Library Prep kit (E7765L) or TruSeq RNA Library Preparation Kit V2 (RS-122-2001) and sequenced with Illumina HiSeq 2500 system. Atx2-wt TRIBE samples were sequenced using HiSeq SBS Kit v4 (FC-401-4003) producing $2 \times 125$ paired-end non-strand-specific reads. TRIBE for all the Atx2 domain mutants were sequenced using HiSeq PE Rapid Cluster Kit v2 (PE-402-4002) to generate $2 \times 100$ paired-end strand-specific data.

## Western blotting

Total protein extracts were prepared from S2 cells as described earlier (*Sudhakaran et al., 2014*). Also, 100 µg protein was loaded for detecting Atx2 and 5 µg for tubulin on 6% and 12% SDS-PAGE gels, respectively, and proteins were transferred to PVDF membrane. The blots were probed using chicken anti-Atx2 (1:1000) and mouse anti-tubulin (1:2000). Corresponding HRP-conjugated secondary antibodies were used at 1:10,000 dilution and developed using SuperSignal West Pico Chemiluminescent Substrate as per the manufacturer's instructions. For detecting V5-tagged ADAR proteins, lysates from fly heads were used as described previously (*Emery, 2007*). Briefly, 10 heads were crushed using plastic pestles in 40 µl extraction buffer (20 mM HEPES pH7.5, 100 mM KCl, 5% glycerol, 10 mM EDTA, 0.1% Triton, 1 mM DTT, 0.5 mM PMSF, 20 mg/ml aprotinin, 5 mg/ml leupeptin, 5 mg/ml pepstatin A). The lysates were cleared by centrifugation at 12,000 *g*, and equal amounts of protein lysates were loaded on a 6% SDS-PAGE gel. Western blots were probed using rabbit anti-V5 (sc83849-R) antibody (1:1000) or mouse anti-tubulin (1:2000) overnight at 4°C. Blots were developed as described above.

## TRIBE data analysis

All sequencing reads obtained post adaptor removal had a mean quality score (Q-Score) >= 37, and so no trimming was required. TRIBE edit details are listed for each experiment in *Supplementary file 2*. The TRIBE data analysis was performed as described previously (*Rahman et al., 2018*) with few modifications. The tools used for analysis are listed in Suppl. Table 6. Briefly, sequencing reads obtained were mapped to dm6 *D. melanogaster* genome using TopHat2 (*Trapnell et al., 2009*) with the parameters '–library-type fr-firststrand -m 1 N 3 –read-edit-dist 3 p 5 g 2 -l 50000 –microexon-search –no-coverage-search -G dm6_genes.gtf'. Non-strand-specific sequencing reads were aligned using tophat2 with the parameters '-m 1 N 3 –read-edit-dist 3 p 5 g 2 -l 50000 –microexon-search –no-coverage-search -G dm6_genes.gtf'. The uniquely mapped SAM output file was loaded in the form of MySQL table with genomic coordinates. Edits for the brain samples were identified by comparing the nucleotide at each position of the genomic coordinates between experiment and control samples, and output was printed as bedgraph file. A threshold file was created by ensuring only edits with coverage of at least 20 reads and 15% edits were retained. This threshold file was used for all further analysis unless specified in the figure legends. All the TRIBE experiments were performed in duplicates, and only the edits identified in both the replicates above the edit threshold are reported.

## S2 cell TRIBE analysis

S2 cells were transfected with 500 ng of *Act-Gal4* and UAS-Atx2-ADAR plasmids (1:1). Cells were harvested 24 hr post transfection, and total RNA was extracted using TRIzol reagent (Invitrogen) as per the manufacturer's protocol for NGS. Illumina libraries preparation and sequencing and TRIBE analysis were performed as described for fly brain samples. S2 cell genomic DNA sequence previously published by *McMahon et al., 2016* was used as control to remove the background edits. S2 cell TRIBE was performed in triplicates, and the edits identified in all the replicates above the edit threshold are reported.

## Differential expression data analysis

RNA sequencing reads were mapped to dm6 *D. melanogaster* genome using STAR v2.5.3a with default parameters, and read counts were obtained using HTseq with '-s reverse' parameter. DeSeq2 was used for differential expression analysis as described previously (*Love et al., 2014*).

## Intron analysis

Nascent transcript analysis was performed by counting reads that emerged from intron sequences as described (*Lee et al., 2020*). Briefly, FASTQ files were mapped to dm6 *D. melanogaster* genome using subjunc with default parameters in the Rsubread software package. Intron annotation SAF file was generated using the scripts found in https://github.com/charitylaw/Intron-reads; (*Lee et al., 2020*). Featurecounts was used to count mapped reads to intron features. Read counts were normalized using DESeq2's median of ratios.

## ARE analysis

ARE*Score* tool (http://arescore.dkfz.de/arescore.pl) was used to perform ARE analysis. Only transcripts with 3'UTR >10 nt in length were considered for analysis. mRNA with highest ARE*Score* was used when multiple transcript variants mapped to the same gene (isoforms of a gene).

## Motif analysis

The edit coordinates from the bed file were extended by 100 bp on either side using Bedtools slop. Intron-less sequences within this ± 100 bp were extracted using twoBitToFa. MEME suite was used to perform motif analysis on the generated FASTA sequences.

## Immunohistochemistry of adult *Drosophila* brains and S2 cells

Six-day-old adult fly brain was dissected in phosphate buffered saline (PBS) and fixed in PBS containing 4% paraformaldehyde (PFA) for 15 min at room temperature. The brains were then processed for immunostaining according to *Sudhakaran et al., 2014*. Atx2 ADARcd was stained using rabbit anti-V5 antibody at 1:200 over one night at 4°C along with neuropil staining using mouse anti-Nc82

(1:100) (*Kittel, 2006*). Secondary antibodies (1:1000) staining was done using anti-rabbit Alexa 488 and anti-mouse Alexa 555 (Molecular Probes) at room temperature for 2 hr. Stained brains were mounted in Vectashield Mounting Medium (Vector Laboratories) and imaged on a Zeiss LSM880 confocal microscope. S2 cells were prepared as described earlier (*Bakthavachalu et al., 2018*). In brief, cells were fixed with 4% paraformaldehyde for 10 min at room temperature, followed by permeabilization with 0.05% Triton-X-100 for 10 min. This was followed by blocking with 1% bovine serum albumin (BSA) for 30 min. The cells were then stained with primary antibodies against Atx2 (1:500) or Rasputin (1:500), followed by probing with corresponding secondary antibodies conjugated with fluorophores. Confocal imaging was done using 60x/1.42 oil objective of Olympus FV3000 microscope. When proteins were SNAP-tagged, SNAP-Surface Alexa Fluor 546 was added after permeabilizing the cells. Confocal images were processed using CellProfiler (https://cellprofiler.org/) to measure granules. At least 80 cells were included in each condition.

### Quantification and statistical analysis
The sample sizes are specified in the figures and figure legends of each experiments. The errors are represented as ± SEM with p-values (*$p<0.05$, ****$p<0.0001$) calculated by two-tailed Student's t-test and Mann–Whitney test. Statistical analysis was performed in GraphPad Prism. Differential expression analysis by DEseq2 is reported for targets with p-value <0.05.

### Contact for reagent and resource sharing
Further information and requests for resources and reagents should be directed to and will be fulfilled by the lead contacts Mani Ramaswami (mani.ramaswami@tcd.ie) and Baskar Bakthavachalu (bbaskar@instem.res.in).

## Acknowledgements
We thank Roy Parker and members of the Ramaswami, VijayRaghavan, and Bakthavachalu labs for useful discussions and/or comments on the manuscript. Thanks to Michael Rosbash for *Drosophila* TRIBE plasmid and to colleagues acknowledged in the Key resources table for generously sharing essential reagents and informal advice. The fly facility at Bangalore Life Science Cluster (BLiSC) provided support with fly stock supply as well as generation of transgenic; CIFF at BLiSC provided essential confocal microscopy support; and Awadhesh Pandit and next-generation genomics facility at BLiSC provided NGS service. We acknowledge *Drosophila* Genomics Resource Centre (supported by NIH grant 2P40OC010949) for *Drosophila* S2 cells.

## Additional information

### Competing interests
Mani Ramaswami: Reviewing editor, *eLife*. K VijayRaghavan: Senior editor, *eLife*. The other authors declare that no competing interests exist.

### Funding

| Funder | Grant reference number | Author |
|---|---|---|
| Science Foundation Ireland | | Mani Ramaswami |
| National Centre for Biological Sciences | NCBS-TIFR Core funding | K VijayRaghavan |
| Science and Engineering Research Board | SB/YS/LS-194/2014 | Baskar Bakthavachalu |
| Indian National Science Academy | INSA/SP/YSP/143/2017 | Amanjot Singh |
| Science and Engineering Research Board | Vajra award | Mani Ramaswami |
| Department of Science and | INSPIRE Fellowship | Khushboo Agrawal |

| | | |
|---|---|---|
| Technology, Ministry of Science and Technology | | |
| Wellcome Trust/DBT India Alliance | IA/1/19/1/504286 | Baskar Bakthavachalu |

The funders had no role in study design, data collection and interpretation, or the decision to submit the work for publication.

## Author contributions
Amanjot Singh, Conceptualization, Data curation, Formal analysis, Funding acquisition, Validation, Investigation, Visualization, Methodology, Writing - original draft, Writing - review and editing; Joern Hulsmeier, Conceptualization, Data curation, Formal analysis, Validation, Investigation, Visualization, Methodology, Writing - original draft, Writing - review and editing; Arvind Reddy Kandi, Data curation, Formal analysis, Validation, Investigation, Visualization, Methodology, Writing - review and editing; Sai Shruti Pothapragada, Jens Hillebrand, Arnas Petrauskas, Validation, Investigation, Visualization, Methodology, Writing - review and editing; Khushboo Agrawal, Data curation, Formal analysis, Validation, Visualization, Methodology, Writing - review and editing; Krishnan RT, Formal analysis, Validation, Investigation, Methodology; Devasena Thiagarajan, Deepa Jayaprakashappa, Data curation, Investigation, Methodology; K VijayRaghavan, Mani Ramaswami, Conceptualization, Resources, Supervision, Funding acquisition, Methodology, Writing - original draft, Writing - review and editing; Baskar Bakthavachalu, Conceptualization, Resources, Data curation, Formal analysis, Supervision, Funding acquisition, Validation, Investigation, Visualization, Methodology, Writing - original draft, Project administration, Writing - review and editing

## Author ORCIDs
Amanjot Singh (iD) https://orcid.org/0000-0001-9793-0404
Joern Hulsmeier (iD) https://orcid.org/0000-0001-9209-5251
Arnas Petrauskas (iD) http://orcid.org/0000-0001-9048-582X
Khushboo Agrawal (iD) http://orcid.org/0000-0001-9159-8615
K VijayRaghavan (iD) http://orcid.org/0000-0002-4705-5629
Mani Ramaswami (iD) https://orcid.org/0000-0001-7631-0468
Baskar Bakthavachalu (iD) https://orcid.org/0000-0002-5114-3429

## Decision letter and Author response
Decision letter https://doi.org/10.7554/eLife.60326.sa1
Author response https://doi.org/10.7554/eLife.60326.sa2

# Additional files

## Supplementary files
• Supplementary file 1. Details of all the primers used in the study.

• Supplementary file 2. All the edits identified (without thresholding) in Targets of RNA-Binding Proteins Identified by Editing experiments are consolidated as a single spreadsheet file. Each genotype is presented as a sheet within this file, and the raw and processed sequencing data is available at GEO accession number GSE153985.

• Transparent reporting form

## Data availability
The RNA Sequencing data has been deposited to GEO under the accession code GSE153985.

The following dataset was generated:

| Author(s) | Year | Dataset title | Dataset URL | Database and Identifier |
|---|---|---|---|---|
| Kandi AR, Ramaswami M, | 2021 | Antagonistic roles for Ataxin-2 structured and disordered | https://www.ncbi.nlm.nih.gov/geo/query/acc. | NCBI Gene Expression Omnibus, GSE153985 |

| | | | |
|---|---|---|---|
| Bakthavachalu B | domains in RNP condensatation | cgi?acc=GSE153985 | |

The following previously published dataset was used:

| Author(s) | Year | Dataset title | Dataset URL | Database and Identifier |
|---|---|---|---|---|
| Rosbash M, McMahon A | 2016 | TRIBE: Hijacking an RNA-editing enzyme to identify cell-specific targets of RNA-binding proteins | https://www.ncbi.nlm. nih.gov/geo/query/acc. cgi?acc=GSM2065948 | NCBI Gene Expression Omnibus, GSM2065948 |

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
