## [Decision Letter]

**Acceptance summary:**

This study offers valuable insight into the Drosophila Ataxin2 protein, examining how it affects translational control and assembly of mRNP granules in the fly brain. Ataxin2 is of broad interest because human mutations in Atx2 cause particular neurodegenerative disorders. The identification of RNA interactions of Atx2 specifically in the RNP granules of neurons in the adult fly will be important for understanding Atx2 function and potentially its pathological roles.

**Decision letter after peer review:**

Thank you for submitting your article "Antagonistic roles for Ataxin-2 structured and disordered domains in RNP condensation" for consideration by *eLife*. Your article has been reviewed by three peer reviewers, and the evaluation has been overseen by Douglas Black as Reviewing Editor and Utpal Banerjee as the Senior Editor. The following individual involved in review of your submission has agreed to reveal their identity: Michael Rosbash (Reviewer #1).

The reviewers have discussed the reviews with one another and the Reviewing Editor has drafted this decision to help you revise the study.

The editors have judged that your manuscript is of interest, but as described below additional experiments and re-review will be required before it can be considered further. We would like to draw your attention to changes in our revision policy that we have made in response to COVID-19 (https://elifesciences.org/articles/57162). First, because many researchers have temporarily lost access to the labs, we will give authors as much time as they need to submit revised manuscripts. We are also offering, if you choose, to post the manuscript to bioRxiv (if it is not already there) along with this decision letter and a formal designation that the manuscript is "in revision at *eLife*". Please let us know if you would like to pursue this option. (If your work is more suitable for medRxiv, you will need to post the preprint yourself, as the mechanisms for us to do so are still in development.)

Summary:

Singh and colleagues examine the interactions of the Ataxin2 protein in *Drosophila*. Ataxin2 is a cytoplasmic RNA binding protein affecting translational control and mRNA stability that mediates assembly of mRNAs into stress granules and neuronal mRNP granules. It is a protein of broad current interest because human mutations in Atx2 can cause spinocerebellar ataxia type II (SCA2) and amyotrophic lateral sclerosis (AML). Both of these diseases seem to involve the aberrant aggregation of RNA binding proteins. In this study the authors examine the RNAs associated with Dm Atx2 specifically in the fly brain, using TRIBE. TRIBE involves fusing a protein of interest to the active domain of adenosine deaminase (ADAR). This fusion protein when expressed in cells will cause adenosine residues adjacent to sites of Atx2 binding to be converted to inosine which codes as guanosine. The presence of A to G transitions in the isolated RNA indicate adjacent binding of Atx2. Expressing their Atx2-ADAR fusion in fly brains with a temperature sensitive Gal4 system, they isolate mRNAs specifically modified by the Atx2 interaction in adult brains. This identified several hundred sites of modification indicating Atx2 interaction with 256 gene transcripts. They show that these transcripts are primarily modified in their 3' UTR sequences and the regions adjacent to these modifications are enriched in AU rich elements (AREs) known to cause mRNA instability. Knocking down Atx2 was found to reduce expression of target mRNA's, consistent with the protein stabilizing RNAs targeted for decay by the AREs. The authors then tested a series of deletions within the fusion protein to examine which domains of Atx2 affect its interaction with these target mRNAs. Surprisingly, they found that the structured domains of Atx2 including the LSM, which are thought to mediate its RNA interaction, had little effect on the identified RNA modifications. Instead removal of a c-terminal intrinsically disordered domain cIDR dramatically reduced the RNA targeting. IDR's are currently being widely studied for their ability mediate the aggregation of RNAs and proteins into intracellular condensates, of which stress granules and neuronal granules are two examples. The authors then test their Atx2 mutants for their intracellular aggregation by transfecting S2 cells with SNAP-tagged Atx2 genes. They find that the ectopic protein does form stress granule-like condensates in cells and this requires the presence of the cIDR. The presence of the LSM domain reduces the size of these condensates.

The reviewers agreed that this study is of substantial potential interest. The identification of RNA interactions of Atx2 that are specifically in the RNA granules of neurons in the adult fly, and different from earlier sets of interactions, is very exciting and will be important for understanding both Atx2 normal function, and potentially its pathological roles. However, all the reviewers found the study to be lacking essential validation, to be missing fundamental controls, and to require an unusually large number of smaller clarifications.

Essential revisions:

1) The loss of mRNAs upon depletion of Atx2 is not rigorously connected to the binding of Atx2 or to ARE's, and it is not necessarily due to increased RNA decay as proposed. In fact, the criteria for inclusion in Figure 3F are not clear, with some fold changes being very small. Do the magnitude of these changes correlate with the ARE score? The connection of Atx2 to these expression changes should be more carefully addressed with reporter constructs in S2 cells. Does Atx-2 expression lead to stabilization of particular target RNAs? Is this effect dependent on the AREs or other elements? Testing the Atx2 mutants, does stabilization or destabilization correlate with granule formation? If the effects on mRNA abundance are directly due to this cytoplasmic protein, then transcription rates should not change. This could be examined by measuring nascent (intron containing) RNA levels.

2) The authors should confirm that each of the proteins with domain deletions exhibit equal steady state expression by immunoblot. This is a standard and essential control as the loss of binding (or gain of more targets) could be due to lesser (or greater) expression of a domain-deleted protein compared to the wt. Similarly, the amount of Atx2 protein remaining after RNAi depletion should be determined by immunoblot. Just measuring the RNA level is not usually considered sufficient.

3) The model proposed in Figure 6 is consistent with their data, but there are other interpretations. One is that the fusion protein is not fully functional. In this regard, they need to show that the TRIBE fusion gene can rescue loss of the wildtype endogenous gene. Second, there remains the possibility that Atx2 has two modes of RNA interaction and that if it engages with RNA through the structured domains, it is in a configuration that does not allow interaction of the deaminase with the target. This would explain why they do not observe previously identified Atx2 interactions in this assay. This should be discussed.

4) It is not clear that the IDR is making a direct interaction with the ARE's. The referenced Biswas paper makes clear that spatial proximity a deaminase bound RNA with an unbound RNA can be sufficient to produce "off target" editing. It is possible that one of the known ARE binding proteins is recruiting the RNAs to the granule where they encounter Atx2. This does not negate the interest of the Atx2 interactions. It may be difficult to test this scenario experimentally, but it should be discussed as a possible, indeed likely, explanation for some of the data.

---

## [Author Response]

Essential revisions:1) The loss of mRNAs upon depletion of Atx2 is not rigorously connected to the binding of Atx2 or to ARE's, and it is not necessarily due to increased RNA decay as proposed. In fact, the criteria for inclusion in Figure 3F are not clear, with some fold changes being very small. Do the magnitude of these changes correlate with the ARE score? The connection of Atx2 to these expression changes should be more carefully addressed with reporter constructs in S2 cells. Does Atx-2 expression lead to stabilization of particular target RNAs? Is this effect dependent on the AREs or other elements? Testing the Atx2 mutants, does stabilization or destabilization correlate with granule formation? If the effects on mRNA abundance are directly due to this cytoplasmic protein, then transcription rates should not change. This could be examined by measuring nascent (intron containing) RNA levels.

We accept the criticism that the proposed role for Atx2 in stabilization of target RNAs via an ARE dependent mechanism could be more rigorously established. We have responded to this concern in two ways: 1A, by providing and highlighting additional lines of evidence in support of our proposal; and 1B, by acknowledging remaining ambiguities: e.g., that our proposal may not be applicable to a randomly selected individual mRNA.

A) Additional lines of evidence in support of Atx2-operating via AREs to stabilize mRNA.

i) The reviewers are correct that AREs scores for individual mRNAs shown in Figure 3F do not have sufficient predictive power. However, a global analysis (Figure 3E) supports a role for Atx2 in stabilizing (or at least increasing steady-state levels) of a significant subset of target mRNAs via an ARE dependent mechanism. This statistically significant, overall correlation between the ARE score of targets and their downregulation after Atx2 k/d is shown in Figure 3E. We have moved the Figure 3F to Figure 3—figure supplement 2 for anyone interested in effects on individual mRNAs.

ii) Our observations in adult fly brain are now supported by additional Atx2 TRIBE analyses we performed in S2 cells. In these cells too, Atx2 -ADAR edits are preferentially seen in AU rich sequences in the 3’UTRs of target mRNAs. As in adult brain, reduction of Atx2 levels using dsRNA decreases steady-state levels of about 49% of identified targets. Interestingly, induction of Atx2-ADAR (which also induces RNP granule formation) resulted in *increased* levels of 74% of the targets downregulated following Atx2 RNAi. These new observations, showing connections between Atx2 and AREs in different cell types, and bidirectional effects of reducing or increasing Atx2(-ADAR) levels (see response to point 3), are added to the revised manuscript as Figure 3—figure supplements 4, 5 and 6 and described in the revised text.

Note that the Atx2 TRIBE targets identified in S2 cells only marginally overlap with the brain targets, which is not surprising considering the mRNA targets from 2 different neuronal subtypes can also be very different (McMahon et al., 2016).

iii) A new analyses of our sequence reads provides further support for a role for Atx2 in regulating mRNA turnover. The changes in steady-state levels of target mRNAs we report could be due to changes either in mRNA stability or transcription. If transcriptional levels were altered, one might predict that Atx2 knockdown would result in altered nuclear-cytoplasmic ratios, reflected in intron-content of target mRNAs. We performed an intron analysis, which showed no significant change in Atx2 target mRNA levels between control and Atx2 RNAi. These new data, included as Figure 3—figure supplement 3 of the revised manuscript, provide some additional support for our proposal that downregulation of Atx2 targets following Atx2 knockdown occurs through a posttranscriptional mechanism, consistent with a role for Atx2 in modulating RNA stabilization.

iv) We have not performed the suggested analyses of individual mRNA reporters in S2 cells for the following reasons. First, we expect some mRNAs to be stabilized, some to be destabilized and some to be unchanged. This is in part from our analysis (previous Figure 3F, now Figure 3—figure supplement 2). But also from known properties of AU-Rich elements, which can be either stabilizing and destabilizing based on the nature of the associated RNP complex (Vasudevan and Peltz, 2001; Otsuka et al., 2019). Thus, the result will not reflect a general role for Atx2, but rather a specific effect on that UTR. We hesitate to select a few targets that would be most likely to fit one model alone. Second, some data on these lines already exists, MS2 tethering assays have already demonstrated that direct binding of Ataxin-2 to the 3’UTR can increase mRNA stability and protein expression in human (Yokoshi et al., Mol Cell, 2017 and Inagaki et al., 2020) as well as in *Drosophila* cells (Lim and Allada, 2013). In these cases Atx2 acts as to increase levels of its target mRNAs, Our published work (e.g. McCann et al., 2011 and Sudhakaran et al., 2014) shows that Ataxin-2 can act a translational repressor, e.g. for CaMKII mRNA and a selection of miRNA target genes.

B) We clarify our model and acknowledge remaining ambiguities: e.g., that our proposal may not be applicable to a randomly selected individual mRNA.

v) In the Abstract we add a small qualifier, now stating saying that Atx2 interactions with AU-rich elements “appear to” modulate stability.

vi) In the Results, we highlight that protein association with AREs can result in either stabilization or destabilization of the mRNA, thereby acknowledging that the effect of high ARE content in a UTR can be hard to interpret without direct experimentation.

vii) The Discussion also acknowledges and considers how the effect of Atx2 on stabilities of individual ARE-containing target mRNAs could vary.

2) The authors should confirm that each of the proteins with domain deletions exhibit equal steady state expression by immunoblot. This is a standard and essential control as the loss of binding (or gain of more targets) could be due to lesser (or greater) expression of a domain-deleted protein compared to the wt. Similarly, the amount of Atx2 protein remaining after RNAi depletion should be determined by immunoblot. Just measuring the RNA level is not usually considered sufficient.

To address the first point, we now include a Western blot (Figure 4—figure supplement 3C) which documents comparable levels of expression across experimental strains expressing different Atx2 domain deletions. These new data show that observed differences in TRIBE targets, particularly for the key lines expressing wild-type, Δ-Lsm and Δ c-IDR forms of Ataxin-2, cannot be accounted for by altered levels of transgene expression.

For the second point, we have also added a Figure 3—figure supplement 1 to show the Atx2 RNAi lines used in the study reduce the Atx2 protein levels when driven in wing discs using patched-Gal4. We have found this difficult to confirm this in Western blots of protein lysates from brains of flies expressing Atx2 RNAi. The difficulty is likely because Atx2 expression is driven by *elav-gal4*, which is variably expressed across neurons in the adult brain and not expressed in glial cells. This probably accounts also for the modest reduction in Atx2 mRNA levels observed in whole brain mRNA analysis. We have altered the Results to indicate this limitation.

However, do note that the additional analyses of the effects of Atx2 knockdown achieved by application of unrelated Atx2 dsRNA constructs on S2 cell TRIBE-targets not only provide data consistent with what we observe with Atx2-RNAi in fly brain but also, by showing the opposing effect of Atx2 overexpression, adds further support for our interpretation of the Atx2 RNAi observations.

3) The model proposed in Figure 6 is consistent with their data, but there are other interpretations. One is that the fusion protein is not fully functional. In this regard, they need to show that the TRIBE fusion gene can rescue loss of the wildtype endogenous gene. Second, there remains the possibility that Atx2 has two modes of RNA interaction and that if it engages with RNA through the structured domains, it is in a configuration that does not allow interaction of the deaminase with the target. This would explain why they do not observe previously identified Atx2 interactions in this assay. This should be discussed.

A) Regarding the full functionality of the Atx2-ADAR fusion protein.

The exact experiment suggested here is difficult to do. Neural overexpression of Atx2 via a UAS-Atx2 causes lethality and no one has yet succeeded in rescuing null ataxin-2 alleles using UAS-Atx2 transgenes, likely because a specific complex promoter is required. However, we offer other lines of evidence showing that C-terminal modifications of Atx2 do not significantly affect Atx2 functions. These observations substantially moderate our concern regarding this issue.

i) We have earlier shown that adding a large c-terminal tag (GFP) to Atx2 using genome engineering at the endogenous locus does not affect Atx2 activity, These Atx2-GFP flies are completely viable, as are similarly engineered flies carrying far more complex C-terminal modifications (-FRT-cIDR-dsRed-STOP FRT-GFP). Similarly, genomic transgenes (containing the native Atx2 promoter, exons, introns and 3’ regulatory sequences) encoding Atx2 carrying a C-terminal GFP tag completely rescue lethality of *atx2* null mutants (Bakthavachalu et al., 2018; Roselli, Bakthavachalu, Ramaswami, unpublished ). Thus, modifying or adding other tags onto the Atx2 C-terminus does not have significant consequences to the protein function.

ii) The addition of a c-terminal SNAP (or GFP) tag on Atx2 does not alter the protein’s native ability to form granules in S2 cells in a manner that is dependent on its cIDR. Atx2-ADAR constructs also behave similarly in S2 cells.

B) Regarding the suggestion that Atx2 may have two modes of RNA interaction

We completely agree that when Ataxin-2 engages with RNA through the Lsm domain, it may be in a configuration that does not allow interaction of the deaminase with its targets. This is the model we favour, but perhaps did not make as clear as we hoped. The point is implicitly acknowledged in the legend of Figure 6. We discuss this further in the Discussion. Moreover, we state in the text that RNA immunoprecipitation or other methodologies may be required identify additional Atx2 targets that are not identified by TRIBE.

4) It is not clear that the IDR is making a direct interaction with the ARE's. The referenced Biswas paper makes clear that spatial proximity a deaminase bound RNA with an unbound RNA can be sufficient to produce "off target" editing. It is possible that one of the known ARE binding proteins is recruiting the RNAs to the granule where they encounter Atx2. This does not negate the interest of the Atx2 interactions. It may be difficult to test this scenario experimentally, but it should be discussed as a possible, indeed likely, explanation for some of the data.

There are two separate issues implicit to this comment. We completely agree that the IDR may not directly contact the ARE and that Atx2 may be placed in proximity to the ARE by association with an independent ARE-binding protein. We now mention this in a revised legend to Figure 6. In one scenario, this indirect engagement with mRNA may occur in cis, i.e. the Atx2-deaminase fusion is brought specifically in contact with the ARE-binding protein and its associated mRNA. In this scenario the mRNA may still a be considered to be a bona fide Atx2 target, just brought to it by an indirect mechanism. In a second scenario, where proximity is achieved simply by concentration of Atx2 and ARE-containing mRNAs in granules, “off-target” editing is certainly possible. To minimize this class of off-target edits, which we expect to occur relatively randomly across the sequence of mRNAs in RNP granules, our data sets only report targets that were edited at the exact same genomic coordinate in independent replicate experiments. Doing so, we might have missed some real targets, but this should minimize mRNAs edited by random deaminase collisions between molecules concentrated in the same granule. We discuss some these issues and in the Discussion and hope that we cite the Biswas paper appropriately.